# Covalent inhibition of endoplasmic reticulum chaperone GRP78 disconnects the transduction of ER stress signals to inflammation and lipid accumulation in diet-induced obese mice

Dan Luo[1†], Ni Fan[1†], Xiuying Zhang[1], Fung Yin Ngo[1], Jia Zhao[1], Wei Zhao[1], Ming Huang[2], Ding Li[3], Yu Wang[2], Jianhui Rong[1]*

[1]School of Chinese Medicine, Li Ka Shing Faculty of Medicine, University of Hong Kong, Hong Kong, China; [2]Department of Pharmacology and Pharmacy, Li Ka Shing Faculty of Medicine, University of Hong Kong, Hong Kong, China; [3]Colleage of Chemistry and Pharmacy, College of Chemistry & Pharmacy, Northwest A&F University, Shaanxi, China

*For correspondence:
jrong@hku.hk

[†]These authors contributed equally to this work

Competing interest: The authors declare that no competing interests exist.

**Abstract** Targeting endoplasmic reticulum (ER) stress, inflammation, and metabolic dysfunctions may halt the pathogenesis of obesity and thereby reduce the prevalence of diabetes, cardiovascular disesases, and cancers. The present study was designed to elucidate the mechnaisms by which plant-derived celastrol ameliorated inflammation and lipid accumulation in obesity. The mouse model of diet-induced obesity was induced by feeding high-fat diet for 3 months and subsequently intervented with celastrol for 21 days. Hepatic and adipose tissues were analyzed for lipid accumulation, macrophage activation, and biomarker expression. As result, celastrol effectively reduced body weight, suppressed ER stress, inflammation, and lipogenesis while promoted hepatic lipolysis. RNA-sequencing revealed that celastrol-loaded nanomicelles restored the expression of 49 genes that regulate ER stress, inflammation, and lipid metabolism. On the other hand, celastrol-PEG4-alkyne was synthesized for identifying celastrol-bound proteins in RAW264.7 macrophages. ER chaperone GRP78 (78 kDa glucose-regulated protein) was identified by proteomics approach for celastrol binding to the residue Cys[41]. Upon binding and conjugation, celastrol diminished the chaperone activity of GRP78 by 130-fold and reduced ER stress in palmitate-challenged cells, while celastrol analog lacking quinone methide failed to exhibit antiobesity effects. Thus, covalent GRP78 inhibition may induce the reprograming of ER signaling, inflammation, and metabolism against diet-induced obesity.

## Editor's evaluation

This paper is of interest to a broad audience of cell biologists, pharmacologists and researchers who work in metabolic diseases. The work provides substantial new insights into the mechanism of action for a plant derived pentacyclic triterpene called celastrol elastrol, in effectively reducing the high fat diet induced tissue hypertrophy in mouse liver and adipose. A series of compelling experiments depict the site of covalent inhibition of the ER stress sensor GRP78 as essential for the beneficial effects in-vivo, supporting the main conclusions.

## Introduction

Dysregulation of cross-talks between endoplasmic reticulum (ER) stress, inflammation, and metabolic pathways hallmark the pathogenesis of obesity, diabetes, atherosclerosis, hypertension, and acute myocardial infarction (*Van Gaal et al., 2006*; *Rocha and Libby, 2009*; *Flegal et al., 2010*). The 78 kDa glucose-regulated protein (GRP78) is an integral ER stress sensor and holds inositol-requiring enzyme 1 (IRE1), protein kinase R-like ER kinase (PERK), and activating transcription factor 6 (ATF6) in the ER lumen to prevent the activation of corresponding downstream substrates under the physiological conditions (*Szegezdi et al., 2006*; *Cnop et al., 2012*). Upon the challenge by stress stimuli, however, unfolded or misfolded proteins are overwhelmingly accumulated in ER compartment, disrupt the complexes of GRP78 with IRE1, PERK, and ATF6, and ultimately induce unfolded protein response (*Cnop et al., 2012*). Obesity and high-fat diet (HFD) not only induce persistent ER stress, leptin resistance, and insulin resistance via activating protein kinase JNK and inducing the phosphorylation of insulin receptor substrate 1 (*Ozcan et al., 2004*; *Ozcan et al., 2009*), but also promote the generation of unfolded or misfolded proteins and the activation of proinflammatory signaling pathways (*Hotamisligil, 2010*; *Pagliassotti et al., 2016*). On the other hand, hyperactive ER stress drives the proinflammatory M1 polarization of adipose tissue macrophages and exacerbates inflammation and metabolic dysfunctions in obesity (*Bujisic and Martinon, 2017*; *Shan et al., 2017*). Importantly, GRP78 is essential for the differentiation of preadipocytes into adipocytes, indicating a critical role in the control of fat contents in body (*Zhu et al., 2013*). The activation of ER stress pathways (e.g., PERK and eIF2α) inversely induces the upregulation of GRP78 expression (*Ozcan et al., 2004*; *Shan et al., 2017*). Thus, GRP78 may be an important molecular target for the development of new therapies against obesity, type 2 diabetes, atherosclerosis, hypertension, and acute myocardial infarction.

Pentacyclic triterpene celastrol as a major bioactive compound from the plant *Tripterygium wilfordii* Hook.f. was recently identified as the best antiobesity drug candidate without evident toxicity at the dose of 100 µg $kg^{-1}$ $d^{-1}$ (*Liu et al., 2015*). Several recent studies suggest that celastrol may induce rapid weight loss through several mechanisms as follows: (1) To enhance leptin activity and reduce food intake in obese mice (*Greenhill, 2015*; *Liu et al., 2015*); (2) To improve insulin sensitivity via inhibiting NF-κB pathway and control the progression of obesity via enhancing antioxidant capacity and lipid metabolism (*Kim et al., 2013*; *Wang et al., 2014*); (3) To activate HSF1–PGC1α transcriptional axis and elicit beneficial metabolic changes against obesity (*Ma et al., 2015*). However, the primary molecular targets have not been well defined for celastrol to bind covalently or noncovalently. At the cell level, celastrol could effectively suppress proinflammatory M1 macrophage polarization and enhance anti-inflammatory M2 macrophage polarization (*Luo et al., 2017*). These results strongly promoted effort to development novel celastrol-based antiobesity drugs. Nevertheless, the pharmacological potential of celastrol is largely limited by its strong hydrophobicity and poor bioavailability. We recently attempted to load celastrol into PEG-PCL nanomicelles to yield Nano-celastrol (*Zhao et al., 2019*). Indeed, Nano-celastrol exhibited highly competitive antiobesity effects and less gastrointestinal retention. These results strongly facilitated the pharmaceutical application of celastrol.

The aim of the present study is to discover the protein targets for celastrol and determine the impact of covalent celastrol–protein conjugation on ER stress in diet-induced obesity. We synthesized celastrol-PEG4-alkyne bearing an alkyne (–C≡C–) group as a molecular probe for the affinity isolation of celastrol-bound proteins. We further determined the in vitro and in vivo effects of celastrol on ER stress, inflammation, and lipid metabolism.

## Results

### Celastrol diminished lipid accumulation in diet-induced obesity

To examine the effects of celastrol on lipid metabolism, we visualized the morphology of adipocytes and the profiles of fatty acids in livers and adipose tissues from diet-induced obese mice. Firstly, liver tissues and adipose tissues were examined by hematoxylin and eosin (H&E) staining. As shown in *Figure 1A, B*, celastrol effectively ameliorated adipose hypertrophy and lipid accumulation in both livers and epididymal fat pads. Secondly, the compositions of fatty acids in liver and adipose tissues were profiled by gas chromatography–mass spectrometry (GC–MS) technology. *Figure 1C, D* shows the GC–MS chromatograms of lipid extracts from the livers and adipose tissues of three treatment groups. Based on the quantitative analysis in *Figure 1E, F*, HFD elevated the contents of several

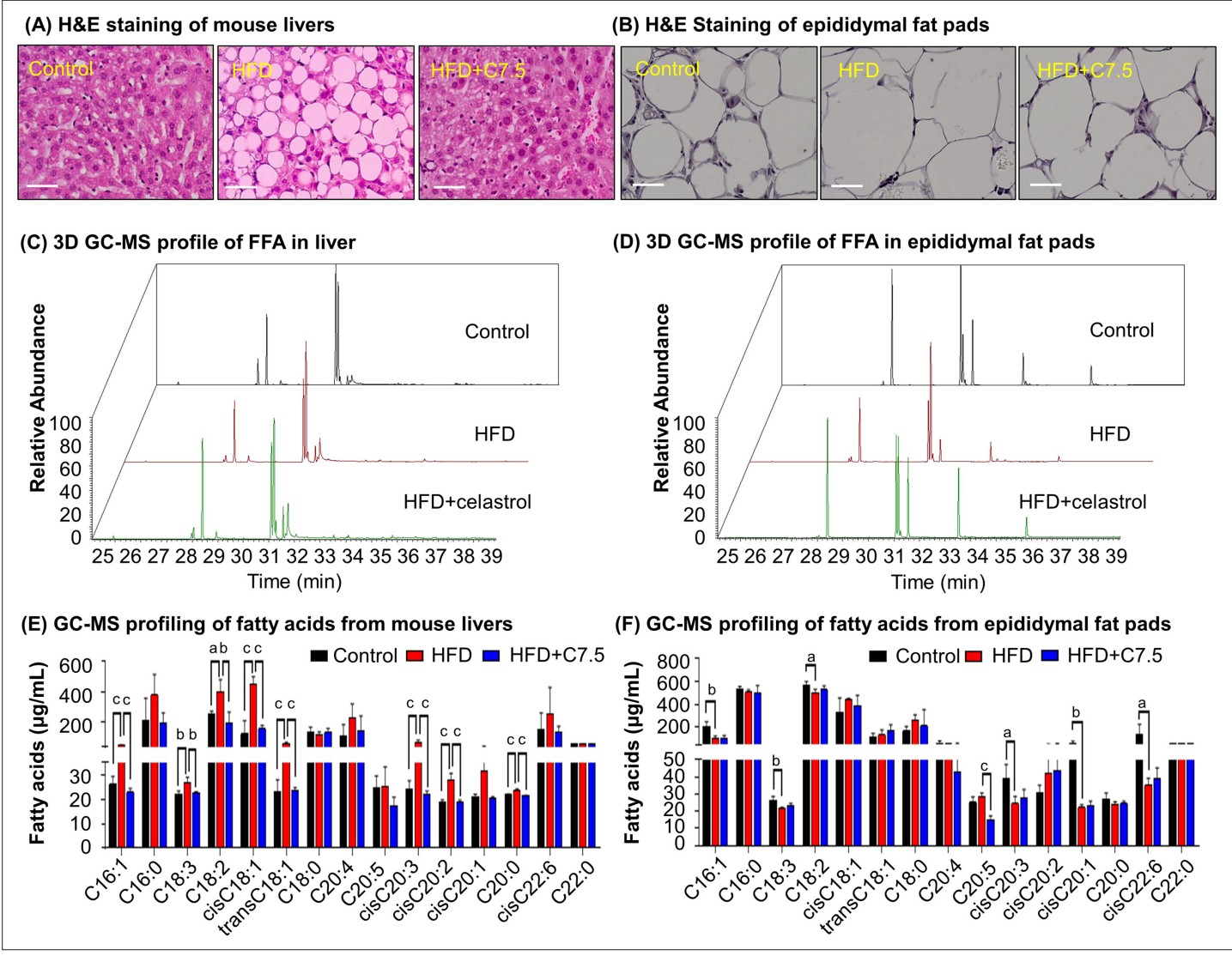

**Figure 1.** Celastrol ameliorates adipose hypertrophy and lipid metabolism. (**A, B**) Hematoxylin and eosin (H&E) staining of mouse hepatic tissues and epididymal fat pads. After 21-day treatment with vehicle or celastrol, the livers and epididymal fat pads were recovered from four groups of C57/BL6 mice (i.e., Control, HFD, HFD + C5, HFD + C7.5) and stained with H&E stain. Representative images were shown. Scale bar represented 34 μm in length. (**C, D**) 3D gas chromatography–mass spectrometry (GC–MS) chromatograms of free fatty acids from murine livers and epididymal fat pads. After 21-day treatment with vehicle or celastrol, livers and epididymal fat pads were recovered from four groups of C57/BL6 mice (i.e., Control, HFD, HFD + C5, HFD + C7.5) for fatty acids extraction and GC–MS profiling. (**E, F**) GC–MS profiling of fatty acids from murine livers and epididymal fat pads. Free and conjugated fatty acids were profiled by GC–MS. Total amounts of individual fatty acids were quantified. The results were presented as mean ± standard deviation (SD) of three independent experiments. a, p < 0.05; b, p < 0.01; c, p < 0.001.

The online version of this article includes the following source data for figure 1:

**Source data 1.** Data for *Figure 1A–D*.

**Source data 2.** Data for *Figure 1E, F*.

polyunsaturated fatty acids (e.g., C16:1, trans-C18:1, cis-C18:1, C18:2, C18:3, cis-C20:2, cis-C20:3) whereas celastrol ameliorated HFD-induced elevations of specific fatty acids in liver (*Figure 1C, E*). By contrast, HFD decreased the contents of long chain polyunsaturated fatty acids eicosoids (e.g., cis-C20:1, cis-C20:3, cis-C22:6) in epididymal fat pads while celastrol showed little activity (*Figure 1D, F*). The results showed that celastrol could diminish lipid accumulation in diet-induced obesity.

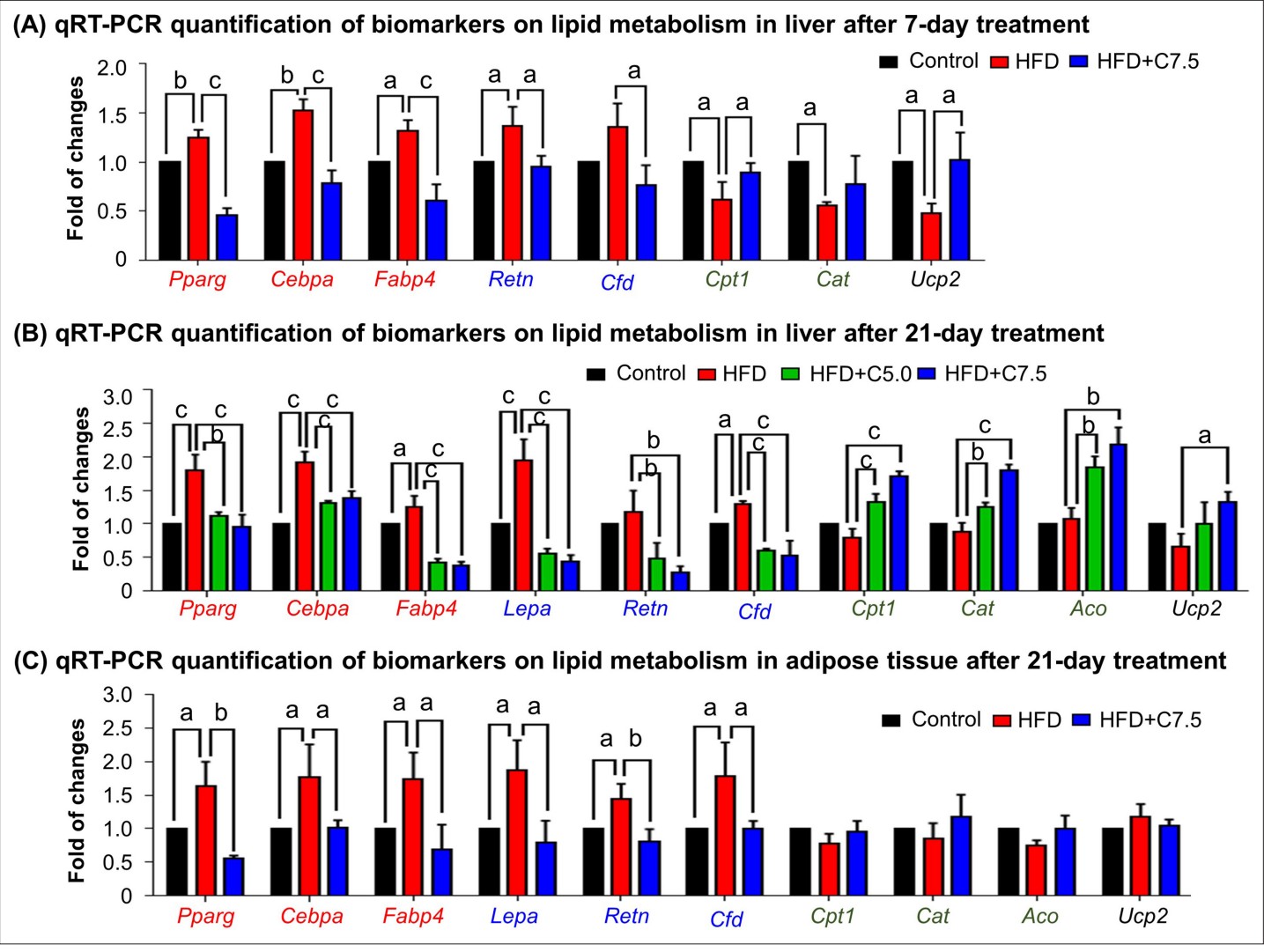

**Figure 2.** Celastrol restores the expression of the biomarkers in lipid metabolism in livers and epididymal adipose tissues. (**A**) One-week effects of celastrol on the biomarkers of lipid metabolism in liver tissues. After 7-day treatment, total RNAs were extracted from livers and analyzed by quantitative real-time PCR (qRT-PCR) technique using Qiagen QuantiTect SYBR Green PCR Kit and specific DNA primers. $N$ = 3. (**B**) Three-week effects of celastrol on the biomarkers in lipid metabolism in liver tissues. After 21-day treatment, total RNAs were extracted from livers and analyzed by qRT-PCR technique using Qiagen QuantiTect SYBR Green PCR Kit and specific DNA primers. $N$ = 3. (**C**) Three-week effects of celastrol the biomarkers in lipid metabolism in epididymal adipose tissues. After 21-day treatment, total RNAs were extracted from epididymal adipose tissues and analyzed by qRT-PCR technique using Qiagen QuantiTect SYBR Green PCR Kit and specific DNA primers. $N$ = 3. HFD, HFD only; C5.0, celastrol (5 mg kg$^{-1}$ d$^{-1}$); C7.5, celastrol (7.5 mg kg$^{-1}$ d$^{-1}$); a, $p < 0.05$; b, $p < 0.01$; c, $p < 0.001$.

The online version of this article includes the following source data for figure 2:

**Source data 1.** Data for *Figure 2A–C*.

## Celastrol inhibited lipogenesis while promoted lipolysis and thermogenesis

To investigate how celastrol regulates lipid metabolism, we employed quantitative real-time PCR (qRT-PCR) technique to determine the expression of the representative biomarkers in lipogenesis, lipolysis, and thermogenesis. As shown in *Figure 2A*, HFD upregulated the expression of adipogenic genes (e.g., *Pparg*, *Cebpa*, *Fabp4*) and adipocyte-secreted hormones (e.g., *Retn*, *Cfd*) while suppressed the lipolytic genes (e.g., *Cpt1*, *Cat*, *Aco*) and thermogenic gene *Ucp2* in liver. After the treatment at the dose of 7.5 mg kg$^{-1}$ d$^{-1}$ for 7 days, celastrol effectively downregulated the expression of the adipocyte-secreted hormones (e.g., *Retn*, *Cfd*), restored the lipolytic genes (e.g., *Cpt1*, *Cat*) and thermogenic gene *Ucp2*, and decreased the expression of the selected adipogenic genes (e.g., *Pparg*,

*Cebpa*, *Fabp4*) to the levels below control. After the animals were treated with celastrol at the doses of 5.0 and 7.5 mg kg$^{-1}$ d$^{-1}$ for 21 days, the levels of these biomarkers in liver and adipose tissues were determined by qRT-PCR technique. As shown in *Figure 2B*, celastrol effectively decreased the upregulation of the lipogenic genes (e.g., *Pparg*, *Cebpa*, *Fabp4*) and adipocyte-secreted hormones (e.g., *Lepa*, *Retn*, *Cfd*) in livers against HFD stimulation. By contrast, celastrol upregulated the lipo-lytic genes (e.g., *Cpt1*, *Cat*, *Aco*) and thermogenic gene *Ucp2* in liver in a dose-dependent manner whereas HFD did not show any effects. As shown in *Figure 2C*, in adipose tissues, celastrol at the dose of 7.5 mg kg$^{-1}$ d$^{-1}$ attenuated the upregulation of the selected lipogenic genes and adipocyte-secreted hormones against HFD stimulation. On the other hand, both HFD and celastrol did not show much effects on the expression of the selected lipolytic and thermogenic genes. These results indi-cated that celastrol inhibited lipogenesis while promoted lipolysis and thermogenesis.

## Celastrol suppressed ER stress signaling in livers and adipose tissues

To validate the in vivo effects of celastrol on the ER stress signaling pathway, firstly, we employed immunofluorescence staining to examine the expression of biomarker GRP78 in hepatic and adipose tissue macrophages from diet-induced obese mice. Mouse liver and epididymal fat pads were recovered from three groups of animals (i.e., Control, HFD, HFD + Celastrol [7.5 mg kg$^{-1}$ d$^{-1}$]), and stained with antibodies against GRP78, whereas macrophages were identified with antibody against biomarker CD68 and the cell nuclei were stained with DAPI. As shown in *Figure 3A, B*, macrophage infiltration and GRP78 expression were greatly enhanced in liver and adipose tissues in obese mice, whereas celastrol treatment effectively inhibited macrophage infiltration and reduced the protein level of GRP78 in macrophages. Secondly, we determined the expression of GRP78 and related ER stress signaling proteins in livers and epididymal fat pads by Western blotting. As shown in *Figure 3C–F*, HFD profoundly increased the protein levels of GRP78, ATF6, and p-eIF2α in livers and epididymal fat pads from obese mice. Celastrol at the dose of 7.5 mg kg$^{-1}$ d$^{-1}$ markedly decreased the protein levels of GRP78, ATF6, and p-eIF2α. These results demonstrated that celastrol effectively suppressed the ER stress signaling pathway in livers and epididymal fat pads.

## RNA-seq transcriptome profiling confirms the effects of celastrol on inflammation, ER stress, and lipid accumulation in liver

To discover the effects of celastrol on hepatic inflammation, ER stress, and lipid metabolism, we deter-mined the effects of Nano-celastrol on the transcriptomic profile in mouse liver by next-generation RNA-seq technology. Based on the expression values and fold of change, a total of 49 differentially expressed genes (DEGs) were initially identified in the livers from the untreated control, HFD, and HFD+ Nano-celastrol groups. As shown in *Figure 4A*, 49 DEGs were compared by hierarchical clus-tering while the effects of HFD and celastrol on gene expression were presented in a heatmap. HFD upregulated nine DEGs including *Apoa4*, *Apoc2*, *Thrsp* while Nano-celastrol suppressed the upreg-ulation of these nine genes and even downregulated three genes (i.e., *Bhmt*, *Car3*, *Mup18*) in HFD-treated mice. By contrast, HFD downregulated 21 DEGs (e.g., *Cyp4a14*, *Mup7*, *Mup14*, *Mup15*) while Nano-celastrol increased the expression of the selected genes to a certain extent and even upreg-ulated four genes (i.e., *Hamp*, *Mt2*, *Igfbp2*, *Serpina1e*) in HFD-treated mice. Among the other 19 genes, Nano-celastrol significantly upregulated four DEGs (i.e., *Lrg1*, *Fgl1*, *Mt1*, *Apom*) and downreg-ulated Scd1 although HFD showed little effect on these genes. Secondly, functional enrichment anal-ysis of DEGs was performed by functional annotation tool DAVID bioinformatics resource 6.8 (https://david.ncifcrf.gov/). As shown in *Figure 4B*, Nano-celastrol significantly affected 6 KEGG pathways, 14 biological processes, and 5 molecular function groups. Interestingly, these functional groups mainly deal with retinal metabolism, lipid metabolism, fatty acid oxidation, cholesterol efflux, glucose metab-olism, stress response, inflammatory response, oxidative stress, detoxification, PPAR pathway, kinase pathways, complement, and coagulation cascade. Thirdly, these DEGs were analyzed for protein–protein interaction networks by online STRING tool (https://string-db.org/). As shown in *Figure 4C*, Nano-celastrol considerably affected 4 networks of gene products including 10 DEGs for Network-1 (*Cyp1a2*, *Cyp2a12*, *Cyp3a25*, *Cyp4a10*, *Cyp4a14*, *Cyp2c44*, *Mdh1*, *Pck1*, *Ugt2b5*, *Aldh1a1*); 8 DEGs for Network-2 (*Apoa4*, *Apoc2*, *Apom*, *C3*, *C8a*, *Creg1*, *Itih3*, *Itih4*); 3 DEGs for Network-3 (*Acat1*, *Ehhadh*, *Hsd17b10*), and 2 DEGs for Network-4 (*Mt1*, *Mt2*). RNA-seq transcriptome profiling confirmed the effects of celastrol on inflammation, ER stress, and lipid accumulation in liver.

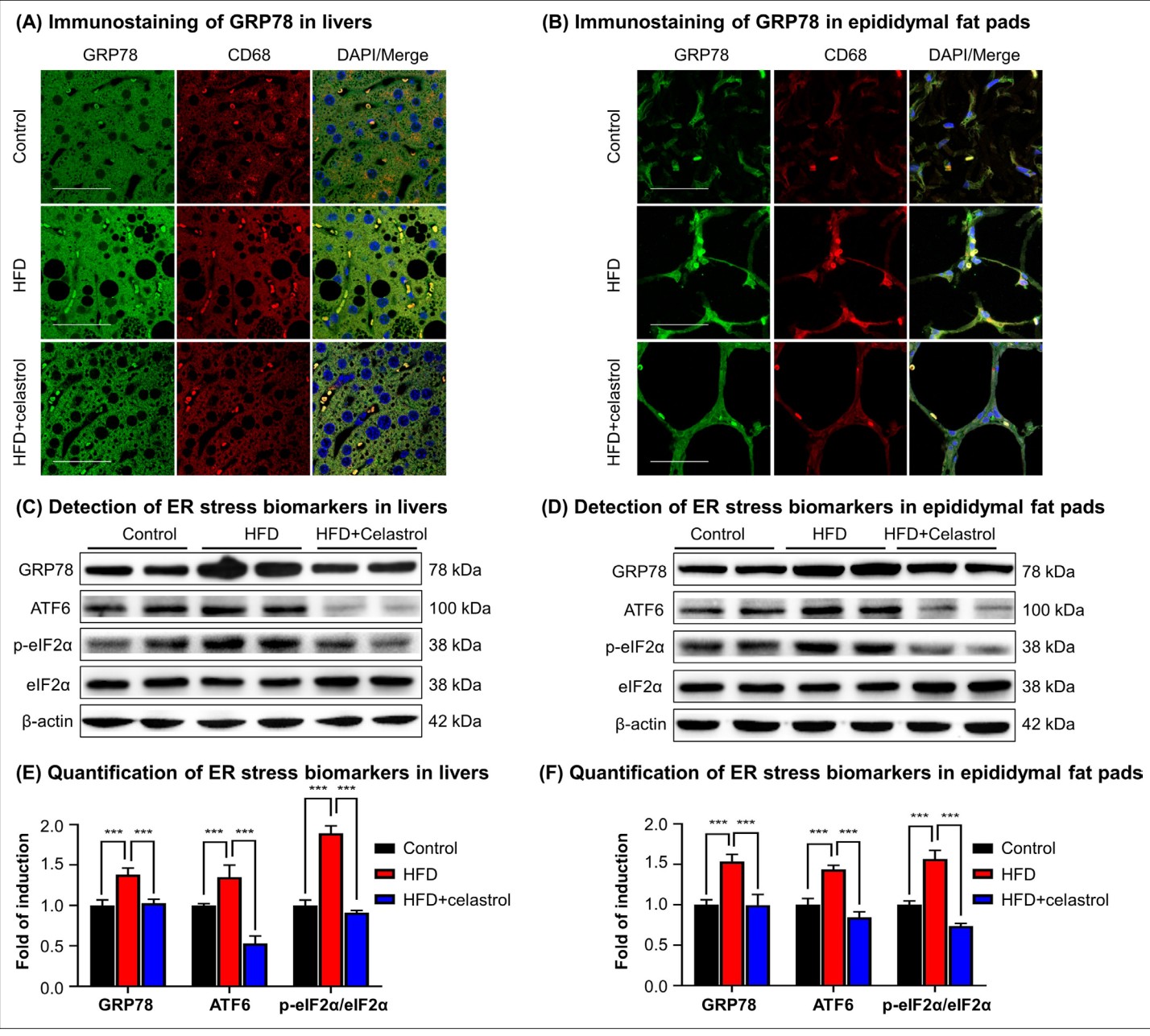

**Figure 3.** Celastrol suppresses endoplasmic reticulum (ER) stress signaling in livers and epididymal adipose tissues. (**A**) Immunostaining of GRP78 in livers. After 21-day treatment, mouse livers were recovered and stained with antibodies against GRP78 and CD68, whereas 2-(4-Amidinophenyl)-6-indolecarbamidine dihydrochloride (DAPI) was used to stain cell nuclei. The sections were imaged under a Zeiss LSM 780 confocal microscope. Representative images were shown. Scale bar, 50 µm. (**B**) Immunosatining of GRP78 in epididymal adipose tissues. After 21-day treatment, mouse epididymal fat pads were recovered and stained with antibodies against GRP78 and CD68, whereas DAPI was used to stain cell nuclei. The sections were imaged under a Zeiss LSM 780 confocal microscope. Representative images were shown. Scale bar, 50 µm. (**C**) Detection of ER stress biomarkers in livers. After 21-day treatment, mouse livers were recovered and analyzed for the expression of ER stress biomarkers by Western blot analysis. Representative blots were shown. (**D**) Detection of ER stress biomarkers in epididymal fat pads. After 21-day treatment, epididymal fat pads were recovered and analyzed for the expression of ER stress biomarkers by Western blot analysis. Representative blots were shown. (**E**) Quantification of ER stress biomarkers in livers. The blots ($n$ = 4) for ER stress biomarkers in livers were quantified by a densitometric method. (**F**) Quantification of ER stress biomarkers in epididymal fat pads. The blots ($n$ = 4) for ER stress biomarkers in epididymal fat pads were quantified by a densitometric method. ***$p < 0.001$.

The online version of this article includes the following source data for figure 3:

**Source data 1.** Data for *Figure 3A*.

*Figure 3 continued on next page*

*Figure 3 continued*

**Source data 2.** Data for *Figure 3B*.

**Source data 3.** Data for *Figure 3C, D*.

## Celastrol ameliorated ER stress in palmitate-challenged macrophages

To verify the effects of celastrol on ER stress, we pretreated RAW264.7 macrophages with celastrol at the concentrations of 0.25, 0.5, 0.75, and 1 µM for 8 hr and incubated with fatty acid palmitate (400 µM) for another 16 hr to induce ER stress. Subsequently, the cellular proteins were extracted and analyzed by Western blotting for the expression of ER stress biomarkers (i.e., GRP78, IRE1α, ATF6, eIF2α, phospho-eIF2α, CHOP, and XBP1). As shown in *Figure 5A, B*, plamitate not only elevated the expressions of ER stress biomarkers (i.e., GRP78, IRE1α, ATF6, CHOP, and XBP1) but also induced

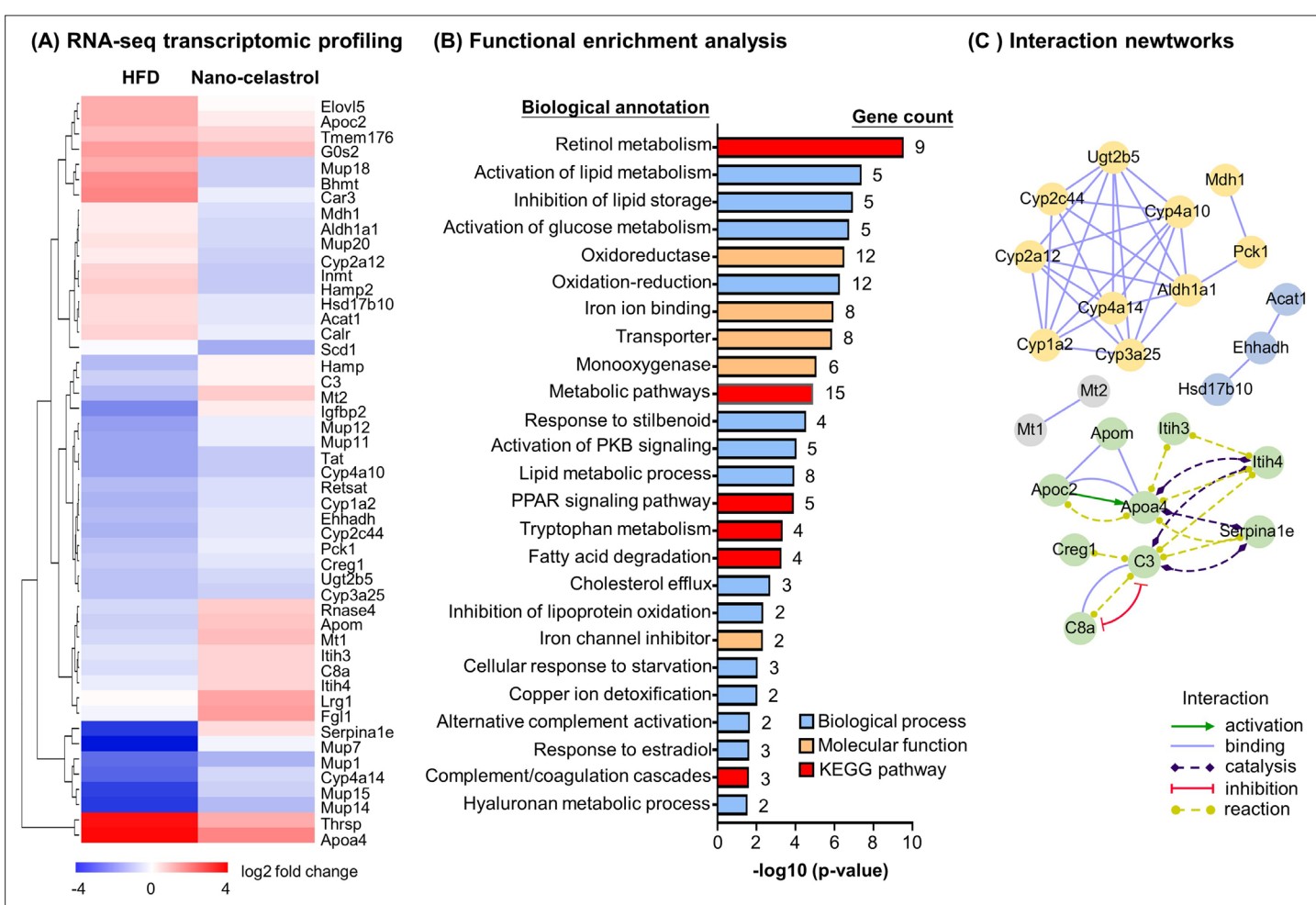

**Figure 4.** RNA-seq transcriptome profiling of celastrol-regulated genes. (**A**) Hierarchical cluster analysis of celastrol-regulated differentially expressed genes (DEGs) in high-fat diet (HFD)-induced obese mice. Expression values are determined by next-generation RNA-seq technology on Illumina HiSeq2500 platform (https://www.genomics.cn/). Data were presented as the expression values (log2 fold change) of the genes in HFD group and HFD + Nano-celastrol group relative to that in Control group. Increases in log2 fold of change are indicated in red hues while decreases are in blue hues. (**B**) Functional enrichment analysis of celastrol-regulated DEGs. Celastrol-regulated DEGs were classified into different functional groups in terms of molecular function, biological process, and Kyoto Encyclopedia of Genes and Genomes (KEGG) pathway using online DAVID v6.8. The number of genes related to each term was shown next to the bar. (**C**) Protein–protein interaction networks of the DEGs. Protein–protein interactions were analyzed by online STRING tool. The interaction networks involving DEGs were represented in different colors and lines using Cytoscape.

The online version of this article includes the following source data for figure 4:

**Source data 1.** Data for *Figure 4A–C*.

**Source data 2.** Differentially expressed genes list for comparison.

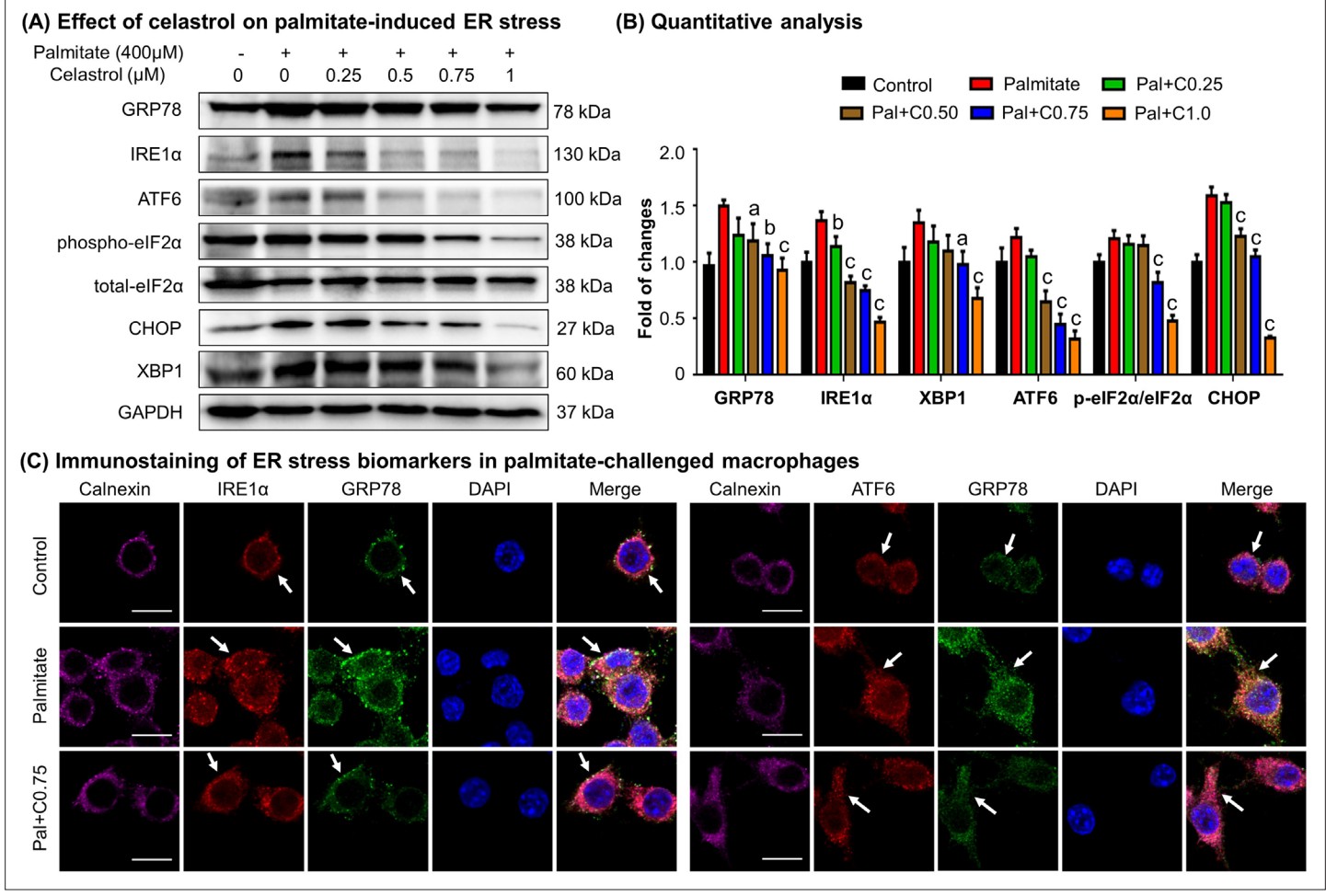

**Figure 5.** Celastrol attenuates palmitate-induced endoplasmic reticulum (ER) stress in RAW264.7 macrophages. (**A**) Western blot analysis for the effects of celastrol on the ER stress biomarkers in palmitate-challenged RAW264.7 cells. RAW264.7 cells were treated with celastrol for 8 hr and subsequently with palmitate for another 16 hr. The cellular proteins were analyzed by Western blotting with specific antibodies. Representative blots were shown. (**B**) Quantitative analysis for the expression levels of ER stress biomarkers. The blots (*n* = 3) were quantified by a densitometric method. C0.25, celastrol (0.25 μM); C0.5, celastrol (0.5 μM); C0.75, celastrol (0.75 μM); C1.0, celastrol (1 μM); a, p < 0.05; b, p < 0.01; c, p < 0.001. (**C**) Colocalization of GRP78, IRE1α, and ATF6 in palmitate-challenged RAW264.7 cells. After the pretreatment with celastrol for 8 hr and subsequently with palmitate for another 16 hr, the cells were stained with antibodies against GRP78, IRE1α, ATF6, and calnexin, whereas DAPI was used to stain the cell nuclei. The cells were imaged under a Zeiss LSM 780 confocal microscope. Representative images were shown. C0.75, celastrol (0.75 μM); palmitate (400 μM). Scale bar, 10 μm.

The online version of this article includes the following source data for figure 5:

**Source data 1.** Original immunoblots.

**Source data 2.** Data for *Figure 5B*.

the phosphorylation of eIF2α. Interestingly, celastrol effectively inhibited the expressions of GRP78, IRE1α, ATF6, CHOP, and XBP1, and attenuated the phosphorylation of eIF2α in a concentration-dependent manner.

To validate the inhibitory effects of celastrol on ER stress, we detected the cellular levels of GRP78, IRE1α, ATF6, and calnexin by immunofluorescence staining. Following the pretreatment with 0.75 μM celastrol for 8 hr and coincubation with palmitate for another 16 hr, RAW264.7 cells were stained with antibodies against GRP78, IRE1α, and ATF6, whereas calnexin was detected as a membrane marker of ER. As shown in *Figure 5C*, palmitate markedly increased the expression of GRP78, IRE1α, and ATF6, while celastrol restored the expression of these proteins to the control levels. Moreover, palmitate caused the dissociation of GRP78 from ER membrane senors IRE1 and ATF6, whereas celastrol

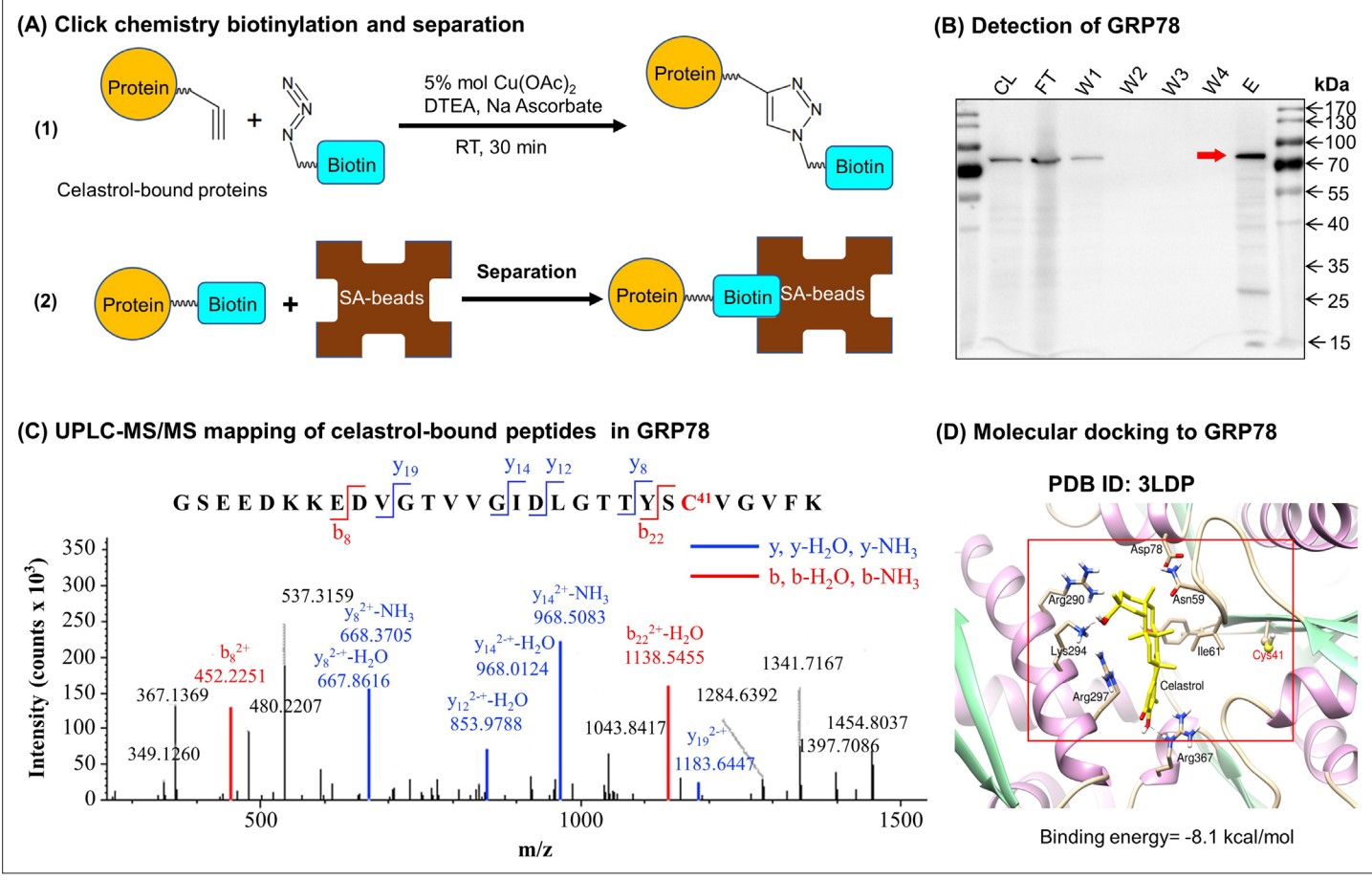

**Figure 6.** Celastrol forms covalent conjugate with endoplasmic reticulum (ER) chaperone GRP78. (**A**) Click chemistry biotinylation and affinity isolation of celastrol-bound proteins. Following 24-hr treatment with celastrol-PEG4-alkyne, the cellular proteins were isolated from RAW264.7 cells and biotinylated with Azide-PEG3-Biotin under Click chemistry conditions. The biotinylated proteins were purified by binding to streptavidin-coated magnetic beads. (**B**) Western blot verification of celastrol-bound proteins. After Click chemistry biotinylation and affinity isolation, celastrol-bound proteins were resolved by 10% sodium dodecyl sulfate–polyacrylamide gel electrophoresis (SDS–PAGE) and visualized by immunoblotting with anti-GRP78 antibody. CL: cell lysate; FT: flow-through; W1: first wash; W2: second wash; W3: third wash; W4: fourth wash; E: elution. (**C**) Mapping of celastrol-binding site in GRP78. Recombinant mouse GRP78 was prepared from DE3 *E. coli* cells and incubated with celastrol in 50 mM sodium bicarbonate (NaHCO₃) buffer containing 10% DMSO. After resolution by Clear-Native-PAGE and Coomassie blue staining, celastrol–GRP78 conjugate was digested with trypsin and analyzed by LC/MS/MS technology. (**D**) Molecular simulation of celastrol–GRP78 interactions. Celastrol was docked into the crystal structure of GRP78 (PDB ID: 3LDP) using Autodock vina in PyRx 0.8 (http://pyrx.sourceforge.net/downloads).

The online version of this article includes the following source data for figure 6:

**Source data 1.** Original immunoblots and figure with the uncropped blots with the relevant bands labeled.

**Source data 2.** Data for *Figure 6C*.

maintained the colocalization of GRP78 with IRE1 and ATF6 in ER compartment. Taken together, these results indicated that celastrol ameliorated ER stress in palmitate-challenged macrophages.

## Identification of celasrol–GRP78 conjugate

To discover the molecular targets of celastrol, we synthesized a new celastrol-PEG4-alkyne bearing an alkyne (–C≡C–) group as a small molecular probe. As shown in *Figure 6A*, following drug treatment, the celastrol-bound proteins were biotinylated through Click chemistry 'azide-alkyne cycloaddition', and separated by affinity enrichment with streptavidin-coated magnetic beads. As shown in *Figure 6B*, ER chaperone GRP78 was identified as a major celastrol-bound protein by mass spectrometry (MS) and verified by Western blotting with anti-GRP78 antibody. The binding of celastrol to GRP78 was characterized by peptide mapping and virtual simulation with AutoDock Vina in PyRx-virtual screen tool package. For peptide mapping, celastrol was incubated with recombinant mouse

GRP78, digested with trypsin and analyzed by liquid Chromatography with tandem mass spectrometry (LC–MS/MS) technology. As shown in *Figure 6C*, celastrol covalently bound to Cys[41] in the peptide with the sequence of GSEEDKKEDVGTVVGIDLGTTYSC[41]VGVFK. As shown in *Figure 6D*, molecular docking also supports that celastrol binds to the Cys[41]-containing binding site in GRP78 structure with the binding energy of −8.1 kcal mol$^{-1}$ and the inhibition constant of 1.55 μM. Therefore, GRP78 was confirmed to be a major celastrol-bound protein.

## Celastrol attenuated the binding of peptide to GRP78

To clarify the effects of celastrol on the chaperone activity of GRP78, we employed surface plasmon resonance (SPR) binding analysis technology to determine the binding of synthetic GRP78-targeting peptide 'WDLAWMFRLPVG' to GRP78 on Biacore X100 SPR analyzer. The affinity was determined by steady-state analysis while Kd values were estimated using Biacore X100 Evaluation Software. Biotinylated GRP78 was initially immobilized onto streptavidin-coated Sensor Chip SA, and synthetic peptide at a wide range of concentrations was subsequently pumped into the system. As shown in *Figure 7A*, the synthetic peptide binds to GRP78 with Kd value of 17.7 nM. As shown in *Figure 7B*, in contrast, when GRP78-coated Sensor Chip was saturated by 10 μM celastrol prior to the binding of GRP78-targeting peptide, the peptide exhibited Kd value of 2.23 μM. These results proved that celastrol could attenuate the binding of peptide to GRP78.

## Quinone methide is required to form covalent celastrol–GRP78 conjugate

To determine the involvement of quinone methide in covalent conjugation, celastrol was reduced by NaBH$_4$ and protected by *O*-methylation to yield celastrol analog (*Figure 7C*), which lacks the chemically reactive quinone methide moiety. When recombinant GRP78 was incubated with excessive celastrol-PEG3-biotin, celastrol analog-PEG3-biotin, or amine-PEG3-biotin, celastrol-PEG3-biotin was readily attached to GRP78 whereas neither of celastrol analog-PEG3-biotin or amine-PEG3-biotin was detectable in Western blot analysis (*Figure 7D*). Free celastrol effectively competed with celastrol-PEG3-biotin for covalent conjugation in a concentration-dependent manner while biotinylation was completely blocked by celastrol at 10-fold excess of molar concentration. On the other hand, the covalent celastrol–GRP78 conjugates in RAW264.7 macrophages were visualized by reacting with a highly reactive probe AFDye555-picolyl azide under Click chemistry conditions as described (*Uttamapinant et al., 2012*). Upon the formation of covalent celastrol–GRP78 conjugate, celastrol-PEG4-alkyne could react with AFDye555-picolyl azide through 'alkyne-azide cycloadduction'. As shown in *Figure 8*, the celastrol-PEG4-alkyne-treated cells showed strong red fluorescence whereas the celastrol-treated cells did not show any red fluorescence. Importantly, the fluorescence label of celastrol-PEG4-alkyne appeared to be well colocalized with GRP78 and calnexin. All these results supported the importance of quinone methide in the formation of covalent celastrol–GRP78 conjugate.

## Quinone methide is required to induce weight loss in diet-induced obese mice

To clarify the importance of covalent GRP78 inhibition in antiobesity effects, the diet-induced obese mice were treated with either celastrol or celastrol analog at the dose of 5 mg kg$^{-1}$ d$^{-1}$ via oral gavage for consecutive 21 days. Based on the measurement of body and body fat contents as shown in *Figure 9A–D*, celastrol effectively reduced the body weight and fat mass while recovered the loss of lean mass in HFD-induced obesity. In contrast, celastrol analog lacking quinone methide moiety failed to ameliorate weight gain, fat accumulation, and lean mass loss. Furthermore, the ipGTT and ITT results in *Figure 9E, F* shows that celastrol analog could no longer ameliorate glucose tolerance and insulin tolerance compared with celastrol. These results proved the critical role of quinone methide in the induction of weight loss in diet-induced obese mice.

## Discussion

Pharmacological modulation of ER stress is a potential therapeutic strategy to treat various metabolic disorders (*Cao and Kaufman, 2013*). Chaperone protein GRP78 is essential to regulate ER stress, inflammation, and lipid metabolism since genetic knockout of GRP78 abolishes the growth,

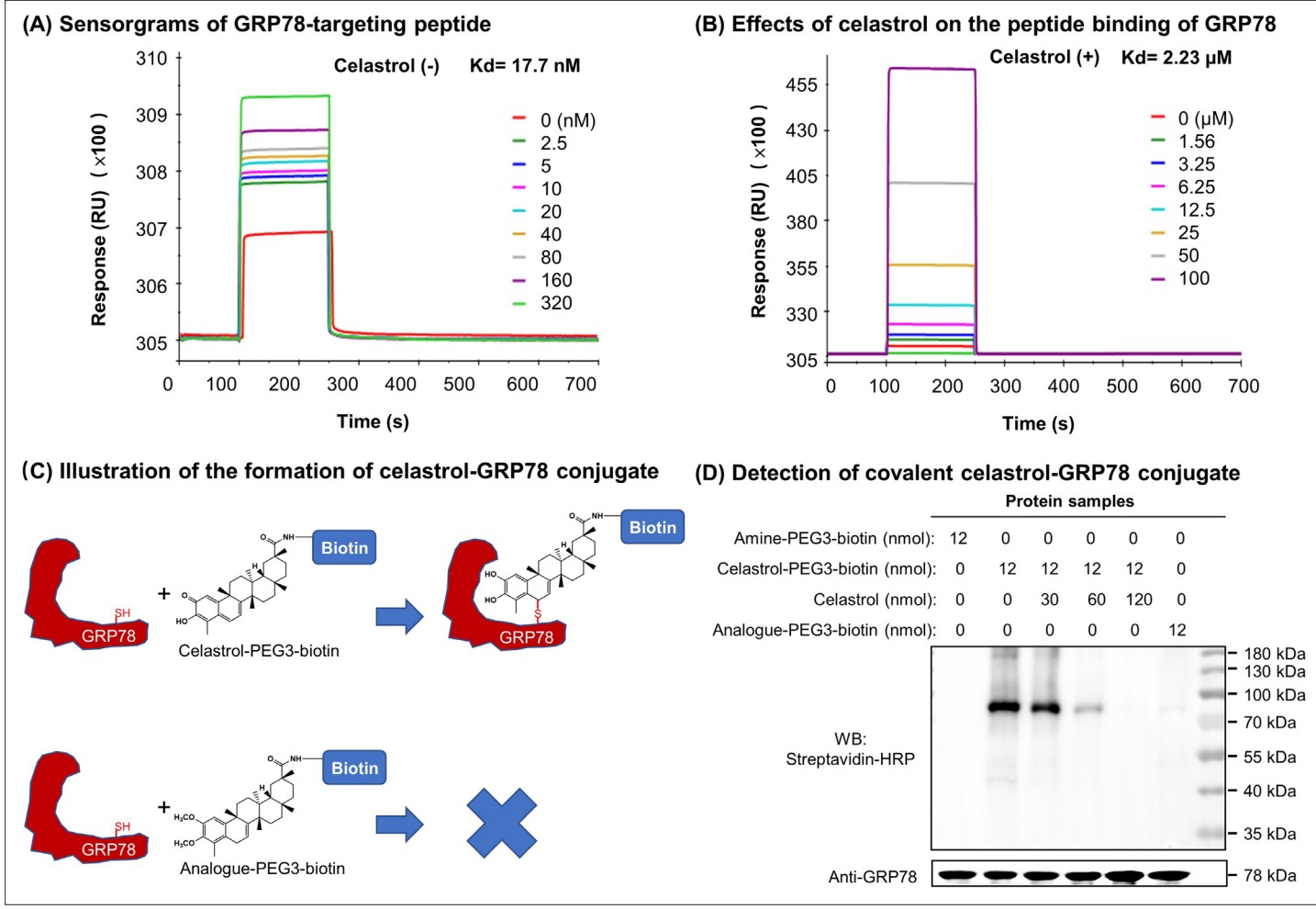

**Figure 7.** Covalent binding of celastrol attenuated the chaperone activity of GRP78 toward peptide. (**A**) Validation of peptide binding to GRP78. After the immobilization of biotinylated GRP78 to the streptavidin-coated Sensor Chip SA, GRP78-targeting peptide (WDLAWMFRLPVG) was pumped into the Biacore X100 SPR system for evaluating the peptide binding to GRP78 under steady-state condition. After four runs, the average Kd value was calculated by Biacore X100 Evaluation Software. Representative sensorgram was shown. (**B**) Effects of celastrol on peptide binding to GRP78. After the saturation with 10 µM celastrol, GRP78-targeting peptide (WDLAWMFRLPVG) (n = 4) was pumped into the Biacore X100 SPR system for calculating the average Kd value for peptide binding to GRP78. Representative sensorgram was shown. (**C**) Scheme illustrating the formation of covalent celastrol–GRP78 conjugate. The cysteine of GRP78 reacts with quinone methide moiety of celastrol whereas the analog lacking quinone methide structure fails to form covalent conjugate with GRP78. (**D**) Detection of covalent celastrol–GRP78 conjugate. Following the incubation of recombinant GRP78 with celastrol-PEG3-biotin or analog-PEG3-biotin, the reaction mixtures were analyzed by Western blotting with streptavidin–HRP or anti-GRP78 antibody.

The online version of this article includes the following source data for figure 7:

**Source data 1.** Data for *Figure 7A, B*.

**Source data 2.** Original immunoblots and figure with the uncropped blots with the relevant bands labelled.

**Source data 3.** Data for the structures, HRMS spectrum, [1]H NMR spectrum, and [13]C NMR spectrum of celastrol analog.

**Source data 4.** Synthesis route of celastrol analog and data for the structures, HRMS spectrum, [1]H NMR spectrum, and [13]C NMR spectrum of intermediates.

**Source data 5.** Data for the structures, HRMS spectrum, [1]H NMR spectrum, and [13]C NMR spectrum of celastrol-PEG3-biotin.

**Source data 6.** Data for the structures, HRMS spectrum, [1]H NMR spectrum, and [13]C NMR spectrum of analog-PEG3-biotin.

differentiation, and maturation of preadipocytes (*Zhu et al., 2013*). By contrast, several small molecule GRP78 inhibitors were recently developed for restoring ER homeostasis, especially, KP1339/ IT-139 was evaluated for anticancer activity in a phase I clinical trial (*Schoenhacker-Alte et al., 2017*; *Chen et al., 2018*; *Gurusinghe et al., 2018*). Other purported GRP78 inhibitors such as Verrucocidin also ameliorate ER stress through different mechanisms other than direct binding to GRP78 (*Thomas*

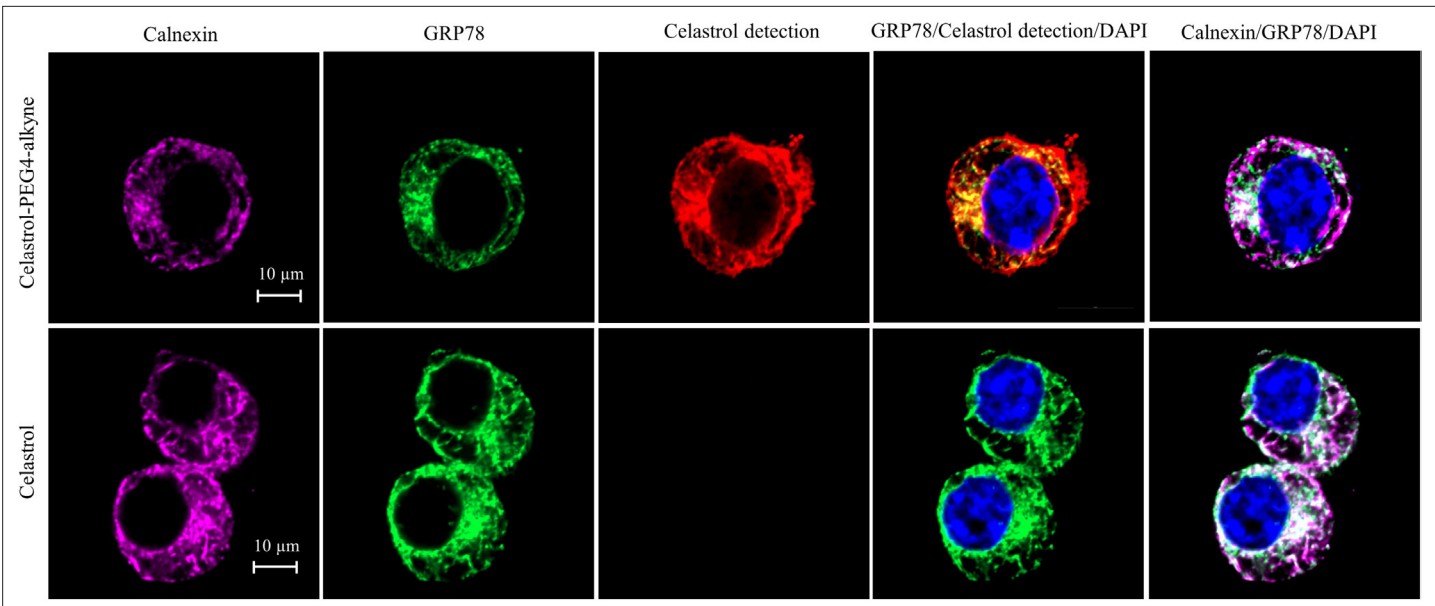

**Figure 8.** Intracellular colocalization of celastrol labeling and GRP78. Following the challenge with palmitate, RAW264.7 cells were treated with celastrol or celastrol-PEG4-alkyne, washed, fixed, reacted with fluorescence dye AFDye555-picolyl azide. The cells were immunostained with antibodies against GRP78 and calnexin, fluorescent secondary antibodies whereas DAPI was used to stain the cell nuclei. The cells were imaged under a Zeiss LSM 780 confocal microscope. Representative images were shown. Scale bar, 10 µm.

The online version of this article includes the following source data for figure 8:

**Source data 1.** Original immunofluorescence staining result of RAW264.7 cells treated with celastrol-PEG4-alkyne.

**Source data 2.** Original immunofluorescence staining result of RAW264.7 cells treated with celastrol.

**Source data 3.** Data for the structures, HRMS spectrum, $^1$H NMR spectrum, and $^{13}$C NMR spectrum of celastrol-PEG4-alkyne.

*et al., 2013*). Currently, little is known about the potential of covalent GRP78 inhibition in the regulation of ER stress. Here, we determined whether celastrol could ameliorate ER stress, inflammation, and metabolic dysfunctions in diet-induced obesity via covalent modification of chaperone GRP78.

HFD is well known to induce the deposition of lipids and alter the composition of fatty acids in liver, adipose tissues, and other organs (*Cameron-Smith et al., 2003*; *Oosterveer et al., 2009*; *Zhao et al., 2019*). Based on transcriptomic profiling and biochemical analysis, HFD upregulated several lipogenic genes for producing fatty acids (*Acc, Fas, Scd1, Elovl6*) and triglycerides (*Dgat, Gpat*), elongase for synthesizing long chain fatty acids such as n-3 and n-6 polyunsaturated fatty acids, and *Srebp1c* and *Pgc1b* for lipid uptake (*Oosterveer et al., 2009*). Our group and others have recently demonstrated that celastrol effectively reduces adipose hypertrophy in mouse and rats models of HFD-induced obesity (*Wang et al., 2014*; *Zhang et al., 2017*; *Zhao et al., 2019*). In the present study, firstly, H&E staining confirmed that celastrol ameliorated adipose hypertrophy and reduced lipid accumulation in liver and epididymal adipose tissues (*Figure 1A, B*). Secondly, GC–MS technology was employed to profile fatty acids in liver and adipose tissues from Control, HFD and HFD + Celastrol groups. Based on the quantitative analysis in *Figure 1E, F*, celastrol ameliorated HFD-induced elevations of various polyunsaturated fatty acids (e.g., C16:1, trans-C18:1, cis-C18:1, C18:2, C18:3, cis-C20:2, cis-C20:3) in liver but showed little activity against HFD-impaired production of long chain polyunsaturated fatty acids eicosoids (e.g., cis-C20:1, cis-C20:3, cis-C22:6) in epididymal fat pads. These results suggest that celastrol may mainly act on the metabolism of fatty acids in liver. Thirdly, qRT-PCR technique was employed to determine the key biomarkers for adipogensis, hormone secretion and energy expenditure. The key findings include: (1) celastrol could effectively antagonize HFD-induced upregulation of adipogenic genes (e.g., *Pparg, Cebpa, Fabp4*) and adipocyte-secreted hormones (e.g., *Lepa, Retn, Cfd*) in livers shortly after 7-day treatment; (2) celastrol markedly elevated the expression of lipolytic genes (e.g., *Cpt1, Cat, Aco*) and thermogenic gene *Ucp2* in a dose-dependent manner (*Figure 2A, B*); (3) celastrol mainly suppressed HFD-induced upregulation of adipogenic genes (e.g., *Pparg, Cebpa, Fabp4*) and adipocyte-secreted hormones (e.g., *Lepa, Retn, Cfd*) in epididymal fat pads (*Figure 2C*).

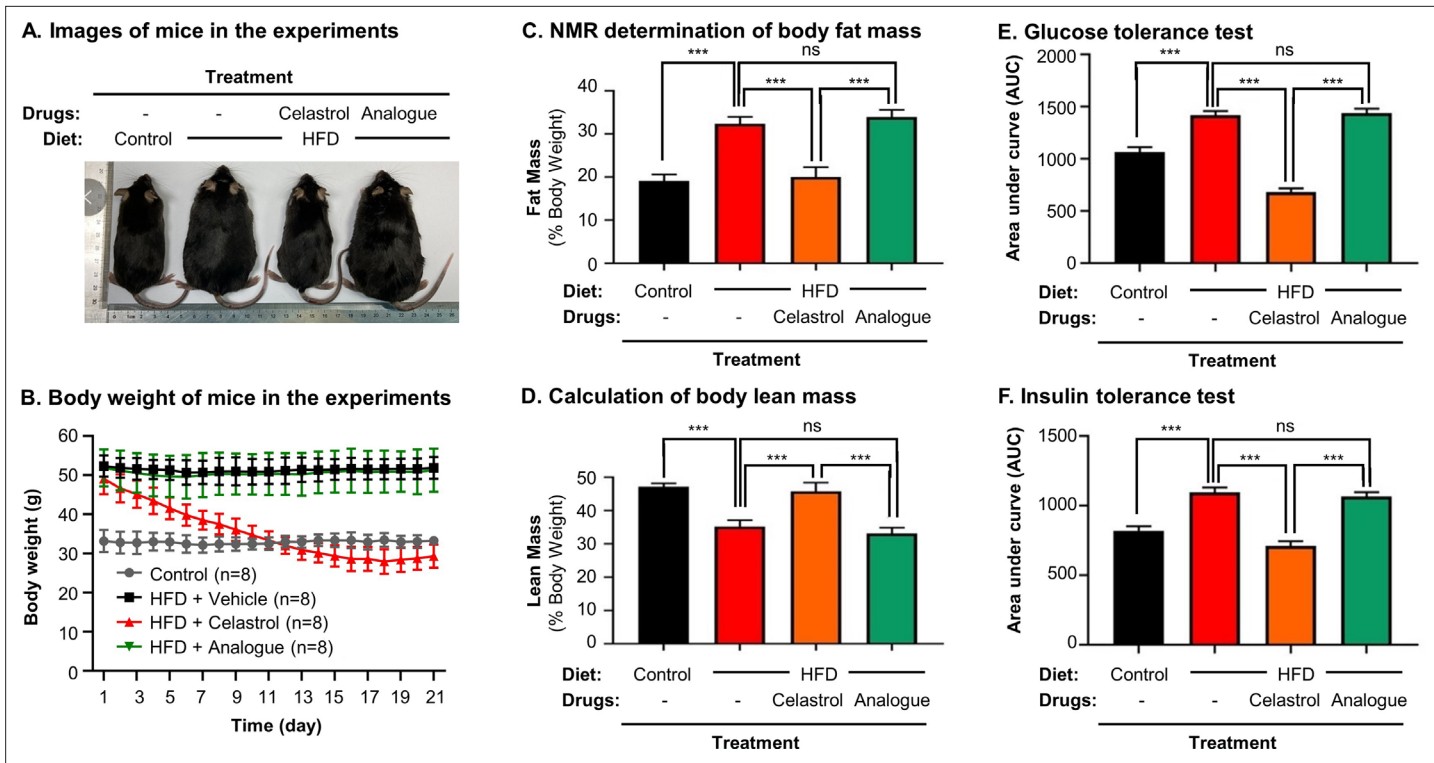

**Figure 9.** Quinone methide is essential to induce weight loss, reduce fat mass, and restore glucose tolerance and insulin tolerance in diet-induced obese C57BL/6N mice. (**A**) Images of experimental mice. Diet-induced obese mice were treated with celastrol or analog at the dose of 5.0 mg kg$^{-1}$ d$^{-1}$ for 21 days, whereas control mice were fed with normal diet. Experimental mice were imaged at day 21. (**B**) Measurement of body weight. Body weight was daily measured (*n* = 8). (**C**) NMR determination of fat contents. After 21-day treatment, fat contents in living mice were analyzed by Bruker minispec NMR analyzer. (**D**) Calculation of body lean mass. Lean mass was calculated from NMR determination of body fat mass as described in (**C**). Body weight was daily measured. (**E**) Glucose tolerance test (GTT). After glucose injection, blood glucose levels were measured and plotted against time. The area under curve (AUC) for each group was calculated. (**F**) Insulin tolerance test (ITT). After insulin injection, blood glucose levels were measured and plotted against time. The AUC for each group was calculated. \*\*\*$p < 0.001$; ns, not significant.

The online version of this article includes the following source data for figure 9:

**Source data 1.** Data for *Figure 9B–F*.

These results suggest that celastrol mainly reprograms the biosynthesis and degradation of fatty acids in liver while inhibits fatty acid biosynthesis and hormone secretion. To facilitate safe control of body weight, we recently prepared Nano-celastrol by entrapping celastrol in PEG-PCL nanomicelles for the treatment of diet-induced obesity (*Zhao et al., 2019*). In the present study, we further identified 49 DEGs in the livers from mice in Control, HFD, and HFD + Nano-celastrol groups by next-generation RNA- sequencing technology (*Figure 4A*). Indeed, *Figure 4B* classified these DEGs into the functional groups of retinol metabolism (i.e., *Aldh1a1*, *Cyp4a10*, *Cyp1a2*, *Cyp2a12*, *Cyp2c44*, *Cyp3a25*, *Cyp4a14*, *Retsat*, *Ugt2b5*), lipid biosynthesis (e.g., *Apoc2*, *C3*, *Cyp1a2*, *Cyp4a10*, *Cyp4a14*, *Ehhadh*, *Elovl5*, *Pck1*, *Scd1*, *Thrsp*), oxidative stress (e.g., *Aldh1a1*, *Creg1*, *Cyp1a2*, *Cyp2a12*, *Cyp3a25*, *Cyp4a10*, *Cyp4a14*, *Ehhadh*, *Hsd17b10*, *Mdh1*, *Retsat*, *Scd1*), lipid degradation (i.e., *Acat1*, *Cyp4a10*, *Cyp4a14*, *Ehhadh*), glucose metabolism (i.e., *Mup1*, *Mup11*, *Mup12*, *Mup18*, *Mup20*), cholesterol efflux (i.e., *Apoa4*, *Apoc2*, *Apom*), and inflammatory response (e.g., *C3*, *C8a*, *Serpina1e*) although some DEGs might be involved in several other biological processes and pathways. By using online software STRING, functional enrichment analysis (*Figure 4C*) revealed four protein–protein interaction networks: (1) lipid and glucose metabolism (i.e., *Mdh1*, *Pck1*, *Aldh1a1*, *Cyp1a2*, *Cyp2a12*, *Cyp2c44*, *Cyp3a25*, *Cyp4a10*, *Cyp4a14*, *Ugt2b5*); (2) apolipoprotein metabolism and inflammatory response (i.e., *Apoa4*, *Apoc2*, *Apom*, *Ltih3*, *Ltih4*, *Serpina1e*, *C3*, *Creg1*, *C8a*); (3) mitochondrial energy homeostasis (i.e., *Acat1*, *Enhadh*, *Hsd17b10*); (4) adipose tissue energy homeostasis (*Mt1*, *Mt2*). Thus, RNA-seq profiling basically confirmed the potential of celastrol in the regulation of oxidative stress, lipid

accumulation, and inflammatory response in the liver. The genes encoding ER stress signaling proteins were not selected probably due to the cutoff of >2.0-fold.

On the other hand, free fatty acids induce ER stress and inflammatory response in macrophages and adipose tissues via TLR-mediated mechanisms (*Han and Kaufman, 2016*). ER stress in turn orchestrates unfolded protein response, inflammation, metabolic dysfunctions, and insulin resistance in obesity and diabetes (*Ozcan et al., 2004*; *Hummasti and Hotamisligil, 2010*). ER chaperone protein GRP78 is integral to the celluluar sensing of fatty acid-induced ER stress and inflammatory stimuli and the regulation of lipid uptake, adipogenesis, lipolysis, glucose metabolism, cholesterol efflux, inflammatory response, and immune response (*Zhu et al., 2013*). GRP78 and CHOP modulate macrophage apoptosis and control the progression of bleomycin-induced pulmonary fibrosis (*Ayaub et al., 2016*). The activation of ER stress sensor IRE1α exacebates obesity-associated inflammation (*Shan et al., 2017*). However, celastrol could reduce ER stress via reducing the protein level of p-PERK/PERK in hypothalamus of diet-induced obese mice (*Liu et al., 2015*). In the present study, the effects of celastrol on the ER stress signaling pathway were examined in the in vitro and in vivo systems. Firstly, livers and epididymal fat pads were collected from experimental mice and examined by immunofluorescence staining and Western blotting. As results, *Figure 3A, B* shows that HFD upregulated GRP78 expression in macrophages whereas celstrol restored the expression level of GRP78. On the other hand, Western blot analysis in *Figure 3C, D* confirmed that HFD upregulated the expression GRP78, ATF6, and p-eIF2α in livers and epididymal fat pads. Celstrol not only restored the expression level of GRP78 but also reduced the levels of ATF6 and p-eIF2α in livers and epididymal fat pads. Secondly, RAW264.7 macrophages were treated by fatty acid palmitate to induce ER stress. Following celastrol treatment, we subsequently employed Western blotting and fluorescence immunostaining to determine the expression levels of ER stress biomarkers. As result, celastrol not only reduced the levels of GRP78, IRE1α, ATF6, CHOP, and XBP1, but also inhibited palmitate-stimulated eIF2α phosphorylation in a concentration-dependent manner (*Figure 5A, B*). Based on the immunostaining results (*Figure 5C*), celastrol decreased the signal intensity of GRP78, ATF6, and IRE1α and preserved the colocalization of GRP78 with ATF6 and IRE1α in the ER compartment. These results indicate that celastrol could effectively suppress the ER stress signaling pathway in the hepatic and adipose tissues.

Celastrol bears a chemically reactive quinone methide moiety and thereby tends to form covalent conjugate with certain proteins such as HIV-Tat, IKKalpha, and beta kinases, Cdc37 and HSF although such covalent attachment appears to be reversible (*Salminen et al., 2010*; *Narayan et al., 2011*; *Zhang et al., 2018*). However, it is critical to identify protein targets with convincing association to ER signaling, anti-inflammatory, and metabolic reprogramming. In the present study, a novel celastrol-PEG4-alkyne bearing an alkyne (–C≡C–) group was firstly synthesized as a small molecule probe through the in-house synthetic procedure (*Cheng et al., 2020*). Secondly, RAW264.7 macrophages were treated with celastrol-PEG4-alkyne and lysed. The cellular proteins were biotinylated via Click chemistry 'azide-alkyne cycloaddition' and separated by affinity enrichment onto streptavidin-coated magnetic beads (*Figure 6A*). The protein fractions were resolved by 10% sodium dodecyl sulfate–polyacrylamide gel electrophoresis (SDS–PAGE) and identified by MALDI-MS technology. The major protein band was identified to be GRP78 and subsequently verified by Western blotting using specific antibody (*Figure 6B*). Thirdly, recombinant mouse GRP78 protein was prepared in *E. coli* BL21 cells and incubated with celastrol. Following trypsin digestion, peptides were mapped by LC–MS/MS analysis. Peptide mapping revealed a celstrol-bound peptide with the sequence of GSEEDKKEDVGT VVGIDLGTTYSC[41]VGVFK (*Figure 6C*). This experiment suggests that celastrol interacts with GRP78 by covalent attachment to Cys[41]. Fourthly, the binding of celastrol to GRP78 was simulated by molecular docking. Interestingly, celastrol binds to the Cys[41]-containing region in GRP78 structure with binding energy of −8.1 kcal mol$^{-1}$ and inhibition constant of 1.55 μM (*Figure 6D*). Fifthly, a recently identified peptide WDLAWMFRLPVG was synthesized to examine the effects of celstrol on the chaperone activity of GRP78 as described (*Arap et al., 2004*). Biacore X100 SPR system was employed to evaluate whether celastrol could alter the binding of peptide WDLAWMFRLPVG to GRP78 on the Sensor Chip. Indeed, celastrol attenuated the chaperone activity of GRP78 since the average Kd value of peptide binding was increased from 17.7 nM to 2.23 μM (*Figure 7A, B*). Sixthly, two new celastrol derivatives were prepared to explore the importance of quinone methide moiety for celastrol to bind to GRP78 (*Figure 7C*). Celastrol-PEG3-biotin was readily detected in association with GRP78 although the binding could be blocked by excess celastrol. By contrast, analog-PEG3-biotin failed

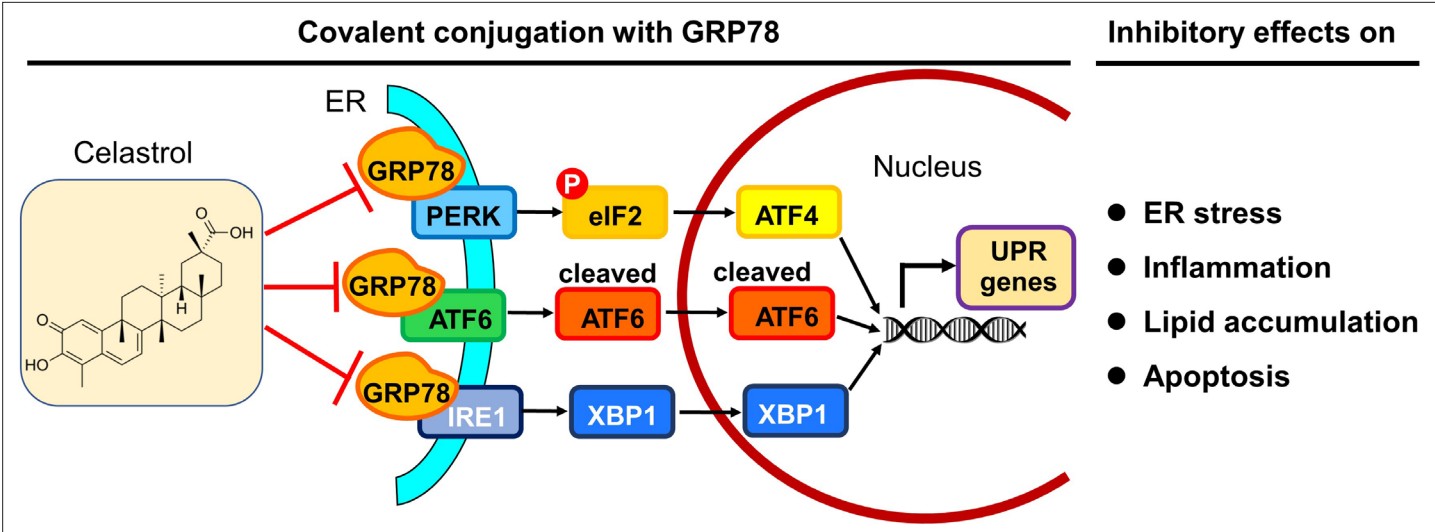

**Figure 10.** Potential mechanisms underlying the activities of celastrol against endoplasmic reticulum (ER) stress, inflammation, and lipid accumulations. The in vitro and in vivo results from the present study suggest that celastrol covalently inhibits the chaperone activity of GRP78 and disconnects the transduction of ER stress signals to downstream inflammatory response and lipid metabolism toward rapid weight loss.

to bind to GRP78 due to the lack of quinone methide moiety (*Figure 7D*). Seventhly, the present study addressed whether the formation of celastrol–GRP78 conjugate could be colocalized with the intracellular GRP78 (*Figure 8*). The cells were treated with celastrol-PEG4-alkyne and subsequently labeled with fluorescence dye AFDye555-picolyl azide whereas GRP78 and another ER biomarker calnexin were detected by immunostaining. As result, the fluorescence of celastrol-PEG4-alkyne labeling was well colocalized with GRP78, and, to a large extent, ER biomarker calnexin. Last but not least, the present study determined whether quinone methide moirty was essential for antiobesity effects (*Figure 9*). Celastrol was reduced by NaBH$_4$ and protected by *O*-methylation to yield celastrol analog. Unlike celastrol, such analog became totally inactive to induce weight loss, reduce fat mass and ameliorate glucose tolerance and insulin tolorance in diet-induced obese mice as assessed by previous described procedure (*Luo et al., 2017*; *Zhao et al., 2019*). These results suggest that the unique quinone methide moiety is critical for celastrol to ameliorate ER stress, inflammation, and lipid accumulation via covalent inhibition of GRP78.

In summary, the present study demonstrated that celastrol ameliorated adipose hypertrophy and reduced lipid accumulation in liver and adipose tissues of diet-induced obese mice. Celastrol might induce rapid reduction of body weight through suppressing lipid uptake, lipogenesis and increasing lipolysis and thermogenesis. As depicted in *Figure 10*, the key finding from the present study was that celastrol covalently inhibited the chaperone activity of GRP78 and could disconnect the transduction of ER stress signals to downstream inflammatory response and lipid metabolism. Therefore, the covalent inhibition of GRP78 is a novel antiobesity mechanism for reprograming the signaling networks that control inflammation, lipid accumulation, ER stress in liver and adipose tissues. Celastrol may be an important lead compound for the development of new selective and irreversible GRP78 inhibitors to treat obesity, diabetes, and related cardiovascular diseases.

## Materials and methods

**Key resources table**

| Reagent type (species) or resource | Designation | Source or reference | Identifiers | Additional information |
|---|---|---|---|---|
| Antibody | Anti-GRP78 (rabbit polyclonal) | Abcam | Cat. #: ab21685 | WB (1:1000) IF (1:500) |
| Antibody | Anti-IRE1α (rabbit monoclonal) | Cell Signaling Technology | Cat. #: 3294 | WB (1:1000) |

*Continued on next page*

*Continued*

| Reagent type (species) or resource | Designation | Source or reference | Identifiers | Additional information |
|---|---|---|---|---|
| Antibody | Anti-XBP1 (rabbit polyclonal) | Abcam | Cat. #: ab37152 | WB (1:1000) |
| Antibody | Anti-Calnexin (goat polyclonal) | Abcam | Cat. #: ab192439 | IF (1:500) |
| Antibody | Anti-ATF-6 (rabbit monoclonal) | Cell Signaling Technology | Cat. #: 65,880 | WB (1:1000) |
| Antibody | Anti-Phospho-eIF2α (rabbit monoclonal) | Cell Signaling Technology | Cat. #: 3398 | WB (1:1000) |
| Antibody | Anti-eIF2α (rabbit monoclonal) | Cell Signaling Technology | Cat. #: 5324 | WB (1:1000) |
| Antibody | Anti-GAPDH (rabbit monoclonal) | Cell Signaling Technology | Cat. #: 2118 | WB (1:1000) |
| Antibody | Anti-β-actin (rabbit polyclonal) | Thermo Fisher Scientific | Cat. #: PA1-183 | WB (1:1000) |
| Antibody | Anti-GRP78 (mouse monoclonal) | Abcam | Cat. #: ab212054 | IF (1:500) |
| Antibody | Anti-CD68 (rabbit polyclonal) | Abcam | Cat. #: ab125212 | IF (1:500) |
| Antibody | Anti-rabbit IgG (H + L) Alexa Fluor 594 (goat polyclonal) | Thermo Fisher Scientific | Cat. #: A-11012 | IF (1:500) |
| Antibody | Anti-mouse IgG (H + L) Alexa Fluor 488 (goat polyclonal) | Thermo Fisher Scientific | Cat. #: A-11001 | IF (1:500) |
| Antibody | Anti-goat IgG H&L Alexa Fluor 647 (donkey polyclonal) | Abcam | Cat. #: ab150131 | IF (1:500) |
| Antibody | Anti-rabbit HRP-conjugated IgG secondary antibody (goat polyclonal) | Sigma-Aldrich | Cat. #: A0545 | WB (1:10,000) |
| Chemical compound, drug | Celastrol | Nanjing Spring and Autumn Biological Engineering Co., Ltd | | Purity >98% (HPLC) |
| Chemical compound, drug | Lipopolysaccharides from *Escherichia coli* O111:B4 | Sigma-Aldrich | Cat #: L4391 | LPS |
| Chemical compound, drug | Sodium palmitate | Sigma-Aldrich | Cat #: P9767 | |
| Chemical compound, drug | Biotin-PEG3-amine | Hunan Huateng Pharmaceutical Company | Cat#: 11,025 | |
| Chemical compound, drug | Alkyne-PEG4-amine | Hunan Huateng Pharmaceutical Company | Cat#: 10,749 | |
| Cell line (*Mus musculus*) | RAW264.7 cells | ATCC | TIB-71 | Authenticated (STR profiling) and mycoplasma test (absent) (Genetic Testing Biotechnology Corporation, Suzhou, China) |
| Commercial assay or kit | TRIzol Reagent | Thermo Fisher Scientific | Cat. #: 15596026 | |
| Commercial assay or kit | RevertAid first-strand cDNA synthesis kit | Thermo Fisher Scientific | Cat. #: K1622 | |
| Commercial assay or kit | SYBR Green mix | QIAGEN | Cat. #: 204,056 | |
| Sequence-based reagent | Mm_Cfd_1_SG | QIAGEN | Cat. #: QT00250495 | *Cfd* |
| Sequence-based reagent | Mm_Fabp4_1_SG | QIAGEN | Cat. #: QT00091532 | *Fabp4* |
| Sequence-based reagent | Mm_Cebpa_1_SG | QIAGEN | Cat. #: QT00311731 | *Cebpa* |

*Continued on next page*

*Continued*

| Reagent type (species) or resource | Designation | Source or reference | Identifiers | Additional information |
|---|---|---|---|---|
| Sequence-based reagent | Mm_Cpt1a_1_SG | QIAGEN | Cat. #: QT00106820 | *Cpt1* |
| Sequence-based reagent | Mm_Lep_1_SG | QIAGEN | Cat. #: QT00164360 | *Lepa* |
| Sequence-based reagent | Mm_Pparg_1_SG | QIAGEN | Cat. #: QT00100296 | *Pparg* |
| Sequence-based reagent | Mm_Retn_1_SG | QIAGEN | Cat. #: QT00093450 | *Retn* |
| Sequence-based reagent | Mm_Ucp2_1_SG | QIAGEN | Cat. #: QT00138943 | *Ucp2* |
| Sequence-based reagent | Mm_Crat_1_SG | QIAGEN | Cat. #: QT00111405 | *Cat* |
| Sequence-based reagent | Mm_Acox1_1_SG | QIAGEN | Cat. #: QT00174342 | *Aco* |
| Sequence-based reagent | Mm_Gapdh_3_SG | QIAGEN | Cat. #: QT01658692 | *Gapdh* |
| Strain, strain background (*Escherichia coli*) | BL21(DE3) | Thermo Fisher Scientific | Cat. #: EC0114 | Competent cells |
| Recombinant DNA reagent | pET-28a | Sigma-Aldrich | Cat. #: 69,864 | |
| Recombinant DNA reagent | pET28a-GRP78 plasmid | This manuscript | | See Materials and methods |
| Software, algorithm | Prism | GraphPad | https://www.graphpad.com/scientific-software/prism/ | |
| Software, algorithm | NIH Image J | NIH | http://imagej.net/ImageJ2 | |

## Chemical synthesis of celastrol derivatives

Celastrol with the purity of over 98% (high-performance liquid chromatography, HPLC) was purchased from Nanjing Spring and Autumn Biological Engineering Co, Ltd. (Nanjing, China). Biotin-PEG3-amine and Alkyne-PEG4-amine were purchased from Hunan Huateng Pharmaceutical Company (Changsha, Hunan, China). The solvents and reagents including 4-methoxybenzyl chloride (PMB-Cl), iodomethane ($CH_3I$), (benzotriazol-1-yloxy)tris(dimethylamino)phosphonium hexafluorophosphate (BOP), sodium bicarbonate ($NaHCO_3$), potassium carbonate ($K_2CO_3$), sodium borohydride ($NaBH_4$), dimethylformamide (DMF), methanol (MeOH), chloroform, dichloromethane (DCM), trifluoroacetic acid (TFA), and triethylamine (TEA) were purchased from Sigma-Aldrich (St Louis, MO, USA). The solvents for extraction and silica gel column chromatography were purchased from RCI Labscan Limited (Bangkok, Thailand) or Duksan Pure Chemicals Company (Ansan, South Korea). All solvents and reagents were used without further preparation unless otherwise indicated. The NMR $^1$H and $^{13}$C spectra were recorded on Bruker AVANCE III HD 400 MHz spectrometer. The chemical shifts were reported in ppm relative to the internal standard signal. The peak splitting patterns were described as follows: s = singlet, d = doublet, t = triplet, q = quartet, m = multiplet, while the coupling constant, *J*, was reported in Hertz unit (Hz). The molecular mass was determined on AB SCIEX X500R Q-TOF mass spectrometer in positive ion mode or negative ion mode. The purity of the products was determined by HPLC analysis on a reversed phase ACE Excel C18 column (2 µm particle size, 100 × 2.1 mm, Advanced Chromatography Technologies Ltd., Scotland) under the control of UltiMate 3000 UPLC system (Thermo Fisher Scientific, USA) equipped with quaternary pump, autosampler, thermostat column compartment, and diode array detection system. The products were generally prepared at the purity of ≥95% based on the integration of UV spectra at 254 nm.

Celastrol analog (2,3-dimethoxydihydrocelastrol) was synthesized from celastrol through several steps of chemical reaction and purified by silica gel column chromatography in the solvents of PE/EA (6:1). Yield: 325 mg, 0.68 mmol, 81.9%. $^1$H NMR (400 MHz, Chloroform-*d*) δ 6.65 (s, 1H), 5.64 (s, 1H), 3.71 (s, 3H), 3.63 (s, 3H), 3.15 (dd, *J* = 20.8, 6.0 Hz, 1H), 2.89 (d, *J* = 20.8 Hz, 1H), 2.32 (d, *J* = 15.4 Hz, 1H), 2.04 (s, 3H), 2.00–1.88 (m, 4H), 1.71 (ddd, *J* = 32.6, 16.0, 10.2 Hz, 2H), 1.57–1.46 (m, 2H), 1.48–1.37 (m, 3H), 1.30 (d, *J* = 15.9 Hz, 2H), 1.21 (s, 3H), 1.11 (s, 3H), 1.07 (s, 3H), 0.95 (s, 3H), 0.79 (d, *J* = 14.8 Hz, 1H), 0.62 (s, 3H). $^{13}$C NMR (101 MHz, CDCl$_3$) δ 181.69, 150.62, 149.22, 144.89, 144.38, 127.63, 125.61, 117.47, 106.08, 60.20, 55.63, 44.24, 43.63, 40.09, 37.45, 37.10, 36.74, 34.67, 34.37, 33.87, 32.71, 31.39, 30.54, 30.41, 30.10, 29.71, 28.77, 27.72, 22.74, 18.32, 11.55. HRMS (*m/z*): [M−H]$^-$ calcd. for C$_{31}$H$_{43}$O$_4$, 479.3161; found, 479.3164.

Celastrol-PEG3-biotin (*N*-(2-(2-(2-((5-(2-oxohexahydro-1*H*-thieno[3,4]-dimidazol-4-yl) pentanamido) ethoxy) ethoxy) ethoxy) ethyl) celastrol-20-carboxamide) was prepared by direct conjugation of celastrol with biotin-PEG3-amine in DMF in the presence of BOP and TEA as previously described (*Cheng et al., 2020*). The product was purified by silica gel column chromatography in the solvents of DCM/MeOH (50:1 to 10:1). Yield: 122 mg, 0.14 mmol, 60.5%. $^1$H NMR (400 MHz, Chloroform-*d*) δ 7.94–7.58 (m, 1H), 7.46–7.28 (m, 1H), 7.01 (d, *J* = 6.6 Hz, 1H), 6.51 (dd, *J* = 12.3, 6.4 Hz, 2H), 6.32 (d, *J* = 7.2 Hz, 1H), 6.01–5.71 (m, 1H), 4.46 (dd, *J* = 8.0, 4.8 Hz, 1H), 4.27 (dd, *J* = 8.0, 4.7 Hz, 1H), 3.62–3.50 (m, 10H), 3.42 (d, *J* = 13.2 Hz, 6H), 3.33–3.25 (m, 2H), 3.09 (q, *J* = 7.3 Hz, 2H), 2.89–2.67 (m, 2H), 2.41 (d, *J* = 15.3 Hz, 1H), 2.19 (d, *J* = 6.2 Hz, 6H), 2.01 (t, *J* = 12.2 Hz, 2H), 1.89–1.77 (m, 3H), 1.67–1.60 (m, 4H), 1.55 (d, *J* = 7.6 Hz, 2H), 1.48 (q, *J* = 7.8, 6.2 Hz, 3H), 1.41 (s, 3H), 1.32 (t, *J* = 7.3 Hz, 2H), 1.23 (s, 3H), 1.12 (s, 3H), 1.08 (s, 3H), 0.60 (s, 3H). $^{13}$C NMR (101 MHz, CDCl$_3$) δ 178.41, 178.29, 173.78, 170.80, 164.99, 164.23, 146.15, 134.54, 127.38, 119.55, 118.17, 117.58, 77.36, 70.43, 70.13, 69.74, 61.96, 60.35, 55.69, 50.72, 46.00, 45.18, 44.46, 43.15, 40.35, 39.46, 39.38, 39.23, 38.30, 36.45, 35.96, 35.06, 33.86, 33.64, 31.73, 31.19, 30.85, 30.09, 29.48, 28.75, 28.32, 28.15, 25.71, 21.80, 18.43, 10.40, 8.73. HRMS (*m/z*): [M+H]$^+$ calcd. for C$_{47}$H$_{71}$N$_4$O$_8$S, 851.4993; found, 851.3923.

Analog-PEG3-biotin (*N*-(2-(2-(2-((5-(2-oxohexahydro-1H-thieno[3,4]-dimidazol-4-yl)pentanamido) ethoxy)ethoxy)ethoxy)ethyl)2,3-dimethoxydihydrocelastrol-20-carboxamide) was prepared by direct conjugation of celastrol with biotin-PEG3-amine in DMF in the presence of BOP and TEA as previously described (*Cheng et al., 2020*). The product was purified by silica gel column chromatography in the solvents of DCM/MeOH (50:1 to 15:1). Yield: 132 mg, 0.15 mmol, 60.0%. $^1$H NMR (400 MHz, Chloroform-*d*) δ 6.86–6.82 (m, 1H), 6.75 (s, 1H), 6.66 (d, *J* = 36.0 Hz, 1H), 6.37 (t, *J* = 5.4 Hz, 1H), 5.74 (dd, *J* = 6.3, 2.1 Hz, 1H), 4.46 (dd, *J* = 7.9, 4.7 Hz, 1H), 4.27 (dd, *J* = 7.9, 4.7 Hz, 1H), 3.83 (s, 3H), 3.74 (s, 3H), 3.60–3.49 (m, 13H), 3.49–3.37 (m, 6H), 3.32–3.26 (m, 2H), 3.10 (d, *J* = 7.0 Hz, 1H), 2.99 (dd, *J* = 20.8, 1.9 Hz, 1H), 2.85 (dd, *J* = 12.8, 4.8 Hz, 1H), 2.71 (d, *J* = 12.7 Hz, 1H), 2.52 (t, *J* = 10.4 Hz, 1H), 2.20 (t, *J* = 7.4 Hz, 3H), 2.14 (s, 3H), 2.08–2.03 (m, 2H), 1.89 (d, *J* = 14.7 Hz, 2H), 1.81–1.75 (m, 1H), 1.70–1.62 (m, 4H), 1.46–1.35 (m, 4H), 1.30 (s, 3H), 1.24 (d, *J* = 3.4 Hz, 1H), 1.20 (s, 3H), 1.13 (s, 3H), 1.08 (s, 3H), 0.96 (d, *J* = 13.6 Hz, 1H), 0.67 (s, 3H). $^{13}$C NMR (101 MHz, CDCl$_3$) δ 178.22, 173.57, 164.18, 150.94, 149.69, 144.79, 144.73, 127.77, 125.54, 117.60, 106.20, 77.36, 70.47, 70.40, 70.14, 70.12, 70.04, 69.74, 61.88, 60.43, 60.35, 55.90, 55.71, 53.56, 44.43, 43.82, 40.52, 39.24, 37.51, 37.27, 36.95, 35.99, 35.03, 34.56, 34.06, 34.00, 31.70, 30.98, 30.76, 30.23, 30.06, 29.00, 28.32, 28.18, 27.97, 25.75, 23.04, 18.45, 11.95. HRMS (*m/z*): [M+H]$^+$ calcd. for C$_{49}$H$_{77}$N$_4$O$_8$S, 881.5462; found, 881.5408.

Celastrol-PEG4-alkyne (*N*-(2-(2-(2-((2-propynyloxy) ethoxy) ethoxy) ethoxy) ethyl) celastrol-20-carboxamide) was generated by direct conjugation of celastrol with alkyne-PEG4-amine in DMF in the presence of BOP and TEA as previously described (*Cheng et al., 2020*). The product was purified by silica gel column chromatography in the solvents of DCM/MeOH (50:1 to 10:1). Yield: 180 mg, 0.27 mmol, 61.6%. $^1$H NMR (400 MHz, Chloroform-*d*) δ 6.99 (d, *J* = 7.1 Hz, 1H), 6.48 (s, 1H), 6.37 (t, *J* = 5.4 Hz, 1H), 6.31 (d, *J* = 7.2 Hz, 1H), 4.17 (d, *J* = 2.4 Hz, 2H), 3.70–3.61 (m, 9H), 3.58 (d, *J* = 4.6 Hz, 2H), 3.53 (dt, *J* = 4.6, 2.9 Hz, 2H), 3.46–3.41 (m, 2H), 3.29 (q, *J* = 5.1 Hz, 2H), 2.46–2.40 (m, 2H), 2.18 (s, 3H), 2.13–2.07 (m, 1H), 1.98 (t, *J* = 13.4 Hz, 2H), 1.89–1.80 (m, 3H), 1.68–1.59 (m, 3H), 1.55–1.49 (m, 2H), 1.40 (s, 3H), 1.22 (s, 3H), 1.12 (s, 3H), 1.09 (s, 3H), 1.01–0.95 (m, 1H), 0.91–0.79 (m, 1H), 0.60 (s, 3H). $^{13}$C NMR (101 MHz, CDCl$_3$) δ 178.35, 178.00, 170.53, 164.87, 146.07, 134.26, 127.40, 119.54, 118.09, 117.21, 79.61, 77.36, 74.77, 70.61, 70.50, 70.42, 70.13, 69.71, 69.08, 58.46, 45.14, 44.45, 43.09, 40.34, 39.42, 39.22, 38.26, 36.44, 35.02, 33.80, 33.61, 31.70, 31.14, 30.86, 30.10, 29.44, 28.74, 21.80, 18.36, 10.34. HRMS (*m/z*): [M+H]$^+$ calcd. for C$_{40}$H$_{58}$NO$_7$, 664.4213; found, 664.4166.

## Animal husbandry and drug treatment

Protocols for animal experiments were approved by the Committee on the Use of Live Animals in Teaching and Research (CULATR No. 3755-15), University of Hong Kong. Obesity was induced in male C57BL/6N mice (age, 3–4 weeks; body weight, 11–13 g) by feeding on 60 kcal% HFD (Research Diets, Inc, New Brunswick, NJ, USA) for 12 weeks before the experiments. Control mice were maintained on chow diet (13.5% from fat calories) (Lab Diet, Inc, St. Louis, MO, USA). Animals were housed under 12 hr of light and 12 hr dark cycle with unrestricted access to food and water at Laboratory Animal Unit, University of Hong Kong. Celastrol was dissolved in saline containing 5% DMSO and 1% Tween-20 prior to administration. For drug treatment, mice were daily administrated with 5 or 7.5 mg kg$^{-1}$ d$^{-1}$ celastrol or celastrol analog via oral gavage for 21 consecutive days. Control mice and HFD mice received vehicle administration similarly by oral gavage. The body weight of mice was daily measured.

## Hematoxylin and eosin (H&E) staining

Following drug treatment, the tissues were examined by H&E staining as previously described (*Cheng et al., 2015*). In brief, the livers and epididymal adipose tissues were collected from diet-induced obese mice after 21-day treatment with celastrol. Tissues were fixed in 10% formalin in phosphate buffer at room temperature for at least 72 hr. After sequential dehydration with 30%, 50%, 70%, 95%, 100% ethanol and xylene, the tissues were embedded in paraffin and dissected into 5 µm tissue sections. Sections were then subjected to H&E staining and examined under light microscope.

## Cell culture and drug treatment

Murine macrophage cell line RAW264.7 cells were obtained from the American Type Culture Collection (Manassas, VA, USA). The cells were cultured in Dulbecco's modified Eagle medium supplemented with 10% fetal bovine serum, and 1% penicillin/streptomycin (Invitrogen, CA, USA) at 37°C in a humidified incubator containing 5% CO$_2$. Binding of palmitate to bovine serum albumin (BSA) was carried out by a modified method as previously reported (*Lu et al., 2013*). Briefly, sodium palmitate was dissolved in 50% phosphate-buffered saline (PBS) and 50% ethanol to make a 100 mM stock solution. Fatty acid-free BSA was dissolved in PBS to make a solution with a concentration of 10 mg ml$^{-1}$. The solutions were filtered through 0.2 µm filters, mixed well to contain 50 mM palmitate, and subjected to two rounds of 10 min incubation at 55°C toward a clear solution for use.

## GC–MS analysis of fatty acids

Following celastrol administration, fatty acids were recovered from livers and epididymal fat pads and quantified as previously described (*Quehenberger et al., 2011*). Briefly, livers (50 mg) and epididymal adipose tissues (50 mg) were homogenized with stainless beads using Tissuelyzer. Organic extraction mixture containing 533 µl chloroform and 267 µl MeOH was added into 200 µl of tissue lysates (4:1, vol/vol) and then incubated on roller at 4°C overnight. After centrifuging at 3000 rpm for 20 min, the lower organic (chloroform/MeOH) fraction was collected. In order to detect free fatty acid, 100 µl of organic fraction was taken to undergo methylation without saponification. The solution was firstly dried under nitrogen (N$_2$) gas and then dissolved in 500 µl of 5% HCl–MeOH solution (wt/vol). After overnight incubation at 50°C, the methylated lipids were extracted three times with 500 µl of isooctane. After dried under nitrogen gas, the extracts were dissolved in 50 µl hexane. In order to detect conjugated fatty acids, 100 µl of organic fraction was subjected to evaporation by nitrogen gas flow and saponification in 500 µl of MeOH–KOH solution at 80°C overnight. At the end of saponification, 50 µl of concentrated HCl was added to adjust pH value to below 5.0. The lipids were extracted three times with 500 µl of hexane, dried by nitrogen gas flow, and methylated in 500 µl of 5% HCl–MeOH solution (wt/vol) at 50°C overnight. The methylated lipids were extracted three times with 500 µl of isooctane, dried under nitrogen gas and dissolved in 50 µl of hexane. Following 1:10 dilution, the samples were analyzed on TRACE 1300 GC-MS system (Thermo Fisher Scientific, MA, USA) equipped with AI1310 autosampler and Thermo TG-5MS column (length, 30 m; internal diameter, 0.25 mm; film, 0.25 µm) (Thermo Fisher Scientific, MA, USA). The analysis employed several internal standards as follows: methyl palmitoleate (C16:1); methyl palmitate (C16:0); methyl γ-linolenate (C18:3); methyl linoleate (C18:2); methyl cis-9-oleic methyl ester (cisC18:1); trans nine elaidic methyl ester (transC18:1); methyl stearate (C18:0); 5,8,11,14-eicosatetraenoic acid methyl ester (C20:4);

5,8,11,14,17-eicosapentaenoic acid methyl ester (C20:5); cis-8,11,14-eicosatrienoic acid methyl ester (cisC20:3); cis-11,14-eicosadienoic acid methyl ester (cisC20:2); cis-11-eicosenoic acid methyl ester (cisC20:1); methyl arachidate (C20:0); cis-4,7,10,13,16,19-docosahexaenoic acid (cisC22:6); and methyl docosanoate (C22:0).

## qRT-PCR determination of mRNA expression

The expression of biomarker mRNAs was determined by qRT-PCR technique as described (*Cheng et al., 2015*). Briefly, the total RNAs were isolated from fresh liver tissues and epididymal adipose tissues using TRIzol reagent (Invitrogen, CA, USA), and converted to the corresponding cDNAs using RevertAid first-strand cDNA synthesis kit (Thermo Fisher, Waltham, MA, USA). qRT--PCR was performed with specific primers from QIAGEN (Valencia, CA, USA) and detection reagent SYBR Green mix (QIAGEN, Valencia, CA, USA). Gene-specific PCR products were subjected to melting curve analysis and quantified by the $2^{-\Delta\Delta Ct}$ method whereas *Gapdh mRNA* was determined as the internal control. The QIAGEN primers were listed as follows: *Cfd* (Mm_Cfd_1_SG; QT00250495), *Fabp4* (Mm_Fabp4_1_SG; QT00091532), *Cebpa* (Mm_Cebpa_1_SG; QT00311731), *Cpt1* (Mm_Cpt1a_1_SG; QT00106820), *Lepa* (Mm_Lep_1_SG; QT00164360), *Pparg* (Mm_Pparg_1_SG; QT00100296), *Retn* (Mm_Retn_1_SG; QT00093450), *Ucp2* (Mm_Ucp2_1_SG; QT00138943), *Cat* (Mm_Crat_1_SG; QT00111405), *Aco* (Mm_Acox1_1_SG; QT00174342), and *Gapdh* (Mm_Gapdh_3_SG; QT01658692).

## RNA-seq identification of DEGs

The next-generation RNA-sequencing technology was employed to determine the effects of Nano-celastrol on HFD-disrupted mRNA expression as previously described (*Trapnell et al., 2012*). In brief, 5 mg of liver tissues were collected from three different mice in each group and homogenized in TRIzol reagent (Thermo Fisher Scientific, Waltham, MA, USA) for the preparation of the total RNAs according to the manufacturer's instruction. The RNA quality was assessed using the Agilent 2100 Bioanalyzer (Agilent technologies, Santa Clara, CA, USA). The ribosome-depleted RNAs and polyA-enriched RNAs were prepared using TruSeq RNA sample prep kit (Illumina, San Diego, CA, USA) and subsequently converted to cDNA libraries with random hexamer primers. The library quality was assessed using the Agilent 2100 Bioanalyzer and quantified by library quantification kit (Illumina, San Diego, CA, USA). RNA-sequencing was performed on Illumina HiSeq2500 platform in BGI Hong Kong (https://www.genomics.cn/). RNA-seq data were aligned to the mouse (mm10) reference genome using STAR aligner (http://code.google.com/p/rna-star/). Differentially expressed genes were selected by DESeq2 (https://bioconductor.org/packages/release/bioc/html/DESeq2.html) using three biological replicates with a nominal significance threshold of p < 0.05 and fold change ≥2. Heatmaps of RNA-seq data were generated using a web-enabled heatmapper (http://heatmapper.ca/). Functional enrichment analysis of DEGs was performed online DAVID 6.8 (https://david.ncifcrf.gov/). Significantly enriched molecular function, biological process, and KEGG pathway of different DEG were selected. The number of genes related to each term was shown next to the bar. Protein–-protein interactions of the DEGs were analyzed by online STRING tool (https://string-db.org/). The molecular networks for DEGs were represented in different colors and lines using Cytoscope 3.7.2 (https://cytoscape.org/).

## Isolation of celastrol-labeled proteins

Following the treatment with celastrol-PEG4-alkyne, the cellular proteins were subjected to Click chemistry biotinylation and proteomic identification as previously described (*Yang et al., 2015*). In brief, RAW264.7 cells were treated with 10 µM celastrol-PEG4-alkyne for 24 hr and lysed with RIPA buffer while iodoacetamide was supplemented to block free cysteine residues in the proteins. The cellular proteins were biotinylated with biotin-azide under Click chemistry conditions (sodium ascorbate, $CuSO_4$, and triazole ligand) to allow 'azide-alkyne cycloaddition' to occur. Biotin-labeled proteins were purified by affinity isolation on streptavidin-coated superparamagnetic beads and identified on ABI4800 MALDI TOF/TOF Analyzer (Applied Biosystems, Foster City, CA) through technical support for proteomics from Center of Genomic Sciences, University of Hong Kong.

## Proteomic identification of celastrol-bound peptides

Full length mouse GRP78 cDNA was cloned by RT-PCR technique using forward primer: 5'-GTCAGGAT CCGAGGAGGACAAGAAGGA-3', reverse primer: 5'-GTCACTCGAGCTACAACTCATCTTTTTCA-3',

and inserted into pET-28a bacterial expression vector after cleavage with restriction enzyme XhoI and BamHI. Competent *E. coli* BL21(DE3) cells were chemically transformed with pET28a-GRP78 plasmid DNA and induced to produce GRP78 protein by 0.2 mM IPTG at 37°C for 4 hr. 6xHis tagged GRP78 was purified by HisTrap HP column (GE Hearlthcare, Uppsala, Sweden) and HiTrap Q HP (GE Health-care, Uppsala, Sweden). The fractions containing GRP78 protein were combined and concentrated by centrifuging through 30 kDa Amicon ultra-15 centrifugal filter units (EMD Millipore, Billerica, MA, USA). A solution of GRP78 protein was prepared at the concentration of 6 mg ml$^{-1}$ in 50 mM NaHCO$_3$ buffer for the binding experiments.

The binding of celastrol to GRP78 was examined at the molar ratio of approximately 1:100 for GRP78 versus celastrol. In brief, 10 µg of purified GRP78 protein was dissolved in 18 µl of 50 mM NaHCO$_3$ buffer and incubated with 6 µg of celastrol in 2 µl DMSO at 4°C overnight and 37°C for 2 hr. The protein complex was resolved by Clear-Native-PAGE (CN-PAGE) and visualized by staining with Coomassie blue. The corresponding protein band was excised as 1 mm pieces. For in-gel digestion, the gel pieces were dehydrated with 50 µl acetonitrile for 15 min and denatured in 100 µl of 8 M urea at room temperature for 1 hr. After the removal of liquid, the gel residues were washed three times with destaining buffer (25 mM ammonium bicarbonate, 50% [vol/vol] acetonitrile) at 37°C for 1 hr and subsequently dehydrated by incubating in 50 µl acetonitrile. The protein complex was digested with sequencing grade trypsin at the enzyme:substrate ratio of 1:50 (wt/wt) in 25 mM NH$_4$HCO$_3$ buffer at 37°C for 18 hr. After desalting with ZipTip C18 tips (Merck Millipore, Billerica, MA, USA), the peptides were dried by lyophilization and resuspended in 3% acetonitrile in 0.1% formic acid.

For LC–MS/MS analysis, peptides were separated on an integrated sparytip column (100 µm i.d.×20 cm) packed with 1.9 µm/120 Å ReproSil-Pur C18 resins (Dr. Maisch GmbH, Germany). The column was eluted with a linear gradient of solvent A (0.1% formic acid in millipore water) and solvent B (0.1% formic acid in acetonitrile) at the flow rate of 250 nl/min as follows: 3–7% B at 0–2 min, 7–22% B at 2–52 min, 22–35% B at 52–62 min, 35–90% B at 62–64 min, 90% B at 64–84 min. Eluted peptides were analyzed with an Orbitrap Fusion mass spectrometer coupled with an Easy-nLC 1000 (Thermo Fisher Scientific, Waltham, MA, USA). Mass spectrometer was operated in positive polarity mode with capillary temperature of 320°C. Full MS survey scan resolution was set to 120,000 with an automatic gain control (AGC) target value of $2 \times 10^5$, maximum ion injection time (IT) of 100 ms, and for a scan range of 350–2000 *m/z*. The MS/MS spectra were acquired in data-dependent mode with a 3 s Top Speed method. Spectra were obtained at 15,000 MS2 resolution with AGC target of 5 $\times 10^4$ and maximum ion injection time (IT) of 40 ms, 1.6 m/z isolation width. Raw mass spectrometry data was processed using Proteome Discover (Thermo Fisher Scientific, Waltham, MA, USA). Raw data was searched against our mouse recombinant GRP78 sequence data. Trypsin/P was specified as a cleavage enzyme allowing up to two missing cleavages. The mass error was set to 10 ppm for precursor ions and 0.02 Da for fragment ions.

## Molecular simulation of celastrol–GRP78 interactions

The crystal structure of human GRP78 (PDB ID: 3LDP) was downloaded from RCSB PDB website (https://www.rcsb.org/). The ligand molecule celastrol (PubChem ID: 122724) was obtained from the PubChem website (https://pubchem.ncbi.nlm.nih.gov/). The protein–ligand docking analysis was carried out by virtual screening software Autodock vina in PyRx 0.8 (https://pyrx.sourceforge.io/) as previously described (*Cheng et al., 2020*). All water molecules were removed from GRP78 protein structure prior to docking. Celstrol molecule was introduced into a 37 × 37 × 35 grid box crossing the protein structure of GRP78, automatically revealing the binding site in the protein. The docking results were analyzed and visualized by PyMOL (https://www.pymol.org/2/) and LigPlus (https://www.ebi.ac.uk/thornton-srv/software/LigPlus/).

## Western blot analysis

The protein levels of specific biomarkers were analyzed by Western blot analysis as previously described (*Cheng et al., 2015*; *Zhao et al., 2017*). The total proteins from liver tissues and epididymal fat pads were extracted in cold RIPA buffer containing protein inhibitor cocktail from Sigma-Aldrich (St. Louis, MO, USA). In brief, 30 µg of the protein samples were resolved by 10% SDS- and subsequently trans-ferred to polyvinylidene difluoride (PVDF) membrane. Following 4 hr incubation in 5% nonfat milk powder or BSA, the membranes were probed with primary antibodies against specific biomarkers

overnight. The bound antibodies were detected with a goat anti-rabbit IgG-HRP conjugate. The blots were visualized by enhanced chemiluminescence (ECL) detection reagents (GE Healthcare, Uppsala, Sweden), and imaged under a BioRad GelDoc imaging system (Hercules, CA, USA). The gel images were analyzed by NIH ImageJ software (http://imagej.net/ImageJ2).

## Immunofluorescence staining

The expression of specific biomarkers in livers and epididymal adipose tissues was examined by immunofluorescence staining as previously described (*Luo et al., 2017*; *Zhao et al., 2019*). In brief, livers and epididymal adipose tissue sections were firstly revitalized in retrieval buffer (pH 6.0), then permeabilized with 0.5% Triton X-100 for 30 min and blocked with 5% normal goat serum in PBS for 2 hr at room temperature. Section slides were then probed with specific primary antibodies at 4°C overnight. After three washes with PBS, the slides were incubated with the fluorescent secondary antibodies (e.g., Alexa Fluor 594-conjugated goat anti-rabbit IgG secondary antibody or Alexa Fluor 488-conjugated goat anti-mouse IgG secondary antibody) for 2 hr at room temperature. The cell nuclei were stained with DAPI. For immunofluorescence staining of the cellular proteins, the cells were sequentially permeabilized, blocked with 5% BSA in PBS and probed with primary antibodies at 4°C overnight. After washes with PBS, cells were incubated with the fluorescent secondary antibodies (e.g., Alexa Fluor 568-conjugated goat anti-rabbit IgG secondary antibody, Alexa Fluor 488-conjugated goat anti-mouse IgG secondary antibody, and Alexa Fluor 647-conjugated donkey anti-goat IgG secondary antibody). The cell nuclei were stained with DAPI. After the removal of excessive fluorescence reagents, the liver and adipose tissues or the cells were imaged on a Zeiss LSM 780 confocal microscope (Carl-Zeiss, Jena, Germany).

## Evaluation of GRP78 chaperone activity by SPR technology

Recombinant 6xHis-tagged human GRP78 protein (Abcam, Cambridge, UK) was biotinylated with EZ-Link Sulfo-NHS-Biotin (Thermo Fisher Scientific, MA, USA) according to the manufacturer's instructions. The biotinylated GRP78 was captured onto a streptavidin-coated Sensor Chip SA (GE Healthcare, NJ, USA) for analyzing the binding of a synthetic GRP78-targeting peptide WDLAWMFRLPVG (GL Biochem Ltd, Shanghai, China) in Biacore X100 SPR system (GE Healthcare, NJ, USA). The system was firstly equilibrated with the running buffer (10 mM 4-(2-Hydroxyethyl)piperazine-1-ethanesulfonic acid (HEPES), 150 mM NaCl, 3 mM ethylenediaminetetraacetic acid (EDTA), 0.005% Tween-20, pH 7.4). The binding of peptide WDLAWMFRLPVG was analyzed at the concentrations from 2.5 nM to 320 µM. In practical, the peptide in the running buffer was loaded to the system at 25°C and a flow rate of 30 µl min$^{-1}$. When 10 µM celastrol was present, the peptide was analyzed at the concentrations from 1.56 to 100 µM. The association phase was set for 180 s, whereas the dissociation phase was set for 300 s. The sensor chip was cleaned by running regeneration buffer (10 mM glycine–HCl, pH 2.2) for 60 s before each new cycle. The affinity was determined by steady-state analysis while the estimation of Kd values was followed the 1:1 binding model with Biacore X100 Evaluation Software.

## Characterization of covalent celastrol–GRP78 conjugate

Recombinant 6xHis-tagged mouse GRP78 was expressed in *E. coli* BL21 cells and purified on a HisTrap HP Nickel column (GE healthcare, Uppsala, Sweden). Twenty micrograms (0.3 nmol) of recombinant GRP78 was dissolved in 18 µl of 50 mM NaHCO$_3$ buffer and incubated with excessive celastrol-PEG3-biotin, celastrol, or celastrol analog-PEG3-biotin at 4°C overnight. The reaction mixtures were precipitated by adding cold acetone and centrifued at 13,000 rpm for 15 min. The protein pellets were dissolved in 50 mM Tris buffer (pH 7.6) containing 500 mM NaCl and 1% Triton-100. An aliquote (5 µg) of protein mixtures was resolved by native SDS–PAGE and subsequently transferred to a PVDF membrane. Following overnight incubation in 5% BSA, the membranes were probed with HRP-conjugated streptavidin (1:10,000) overnight. The blots were visualized by ECL detection reagents under a BioRad Gel Doc imaging system. The GRP78 was detected with anti-GRP78 antibody as index of protein loading.

## Detection of intracellular covalent celastrol–GRP78 conjugate

The formation of covalent celastrol–GRP78 conjugate was examined in RAW264.7 macrophages as previously described (*Uttamapinant et al., 2012*). Briefly, RAW264.7 cells were sequentially treated

with 400 µM palmitate for 16 hr, and 1 µM celastrol-PEG4-alkyne or celastrol for another 2 hr. The cells were fixed with 4% formaldehyde in PBS (pH 7.4), permeabilized with 0.5% Triton X-100 in PBS for 30 min. Click chemistry reaction was performed by the incubation in the dark with 10 µM AFDye555-picolyl azide, 8 mM $CuSO_4$, 8 mM BTTAA, and 500 mM sodium ascorbate in PBS at room temperature for 0.5 hr. After washing with PBS for three times and blocking with 5% BSA, the cells were subjected to immunostaining for GRP78 and calnexin with specific primary antibodies and fluorescent secondary antibodies. The cell nuclei were stained with DAPI. The cells were washed with PBS for three times and imaged under a Zeiss LSM 900 confocal microscopy (Carl-Zeiss, Jena, Germany).

## NMR determination of fat mass

The body fat contents of mice were assessed by a benchtop Bruker minispec LF90 TD-NMR analyzer from Bruker Optics Inc (Billerica, MA, USA) essentially as previously described (*Luo et al., 2017*). In brief, the diet-induced obese mice were treated with celastrol and celastrol analog at the dose of 5 mg $kg^{-1}$ $d^{-1}$ via oral gavage for consecutive 21 days. For the determination of fat mass, the benchtop minispec instrument was firstly calibrated using Bruker standards. Mice were weighted and subsequently placed into the instrument for noninvasive determination.

## Assays of glucose tolerance and insulin sensitivity

Glucose tolerance and insulin sensitivity were assayed as described (*Luo et al., 2017*). For ipGTT, mice were fasted for 16 hr (18:00 p.m. to 10:00 a.m.) and then received d-glucose at the dose of 1 $gkg^{-1}$. Blood samples were collected from mouse tail vein at 0, 15, 30, 60, 90, and 120 min, and measured for blood glucose levels using glucose test paper. For ITT, mice were starved for 6 hr (10:00 a.m. to 16:00 p.m.) and then received insulin at the dose of 0.75 IU $kg^{-1}$. Blood samples were collected from mouse tail vein at 0, 15, 30, 60, 90, and 120 min, and measured for blood glucose levels using glucose test paper.

## Statistical analysis

The results were presented as mean ± standard deviation (SD). The differences between two groups were analyzed by one-way analysis of variance with Dunnett's post hoc test using GraphPad Prism software (La Jolla, CA, USA). The p value of less than 0.05 was considered as statistically significant.

## Acknowledgements

The authors are grateful to Ms. Lin Lin from SUSTech Core Research Facilities of Southern University of Science and Technology (Shenzhen, China) for her technical assistance in proteomic identification. The authors are grateful to Mr. Zhao Chenliang from Hong Kong Baptist University for his help in recording NMR [1]H and [13]C spectra of compounds. This work was supported by General Research Fund (GRF) grants (17146216, 17100317, 17119619) from the Research Grants Council of Hong Kong, the National Natural Science Foundation of China (21778046), and the Seed Funding Programme for Basic Research (201611159156) from the University of Hong Kong.

## Additional information

### Funding

| Funder | Grant reference number | Author |
| --- | --- | --- |
| Research Grants Council, University Grants Committee | 17146216 | Jianhui Rong |
| National Natural Science Foundation of China | 21778046 | Jianhui Rong |
| University of Hong Kong | 201611159156 | Jianhui Rong |

| Funder | Grant reference number | Author |
| --- | --- | --- |
| Research Grants Council, University Grants Committee | 17100317 | Jianhui Rong |
| Research Grants Council, University Grants Committee | 17119619 | Jianhui Rong |

The funders had no role in study design, data collection, and interpretation, or the decision to submit the work for publication.

## Author contributions

Dan Luo, Conceptualization, Data curation, Investigation, Methodology, Writing – original draft; Ni Fan, Conceptualization, Data curation, Formal analysis, Investigation, Methodology, Writing – original draft, Writing - review and editing; Xiuying Zhang, Ming Huang, Ding Li, Investigation; Fung Yin Ngo, Jia Zhao, Wei Zhao, Yu Wang, Data curation; Jianhui Rong, Conceptualization, Project administration, Supervision, Writing - review and editing

## Author ORCIDs

Jianhui Rong (ID) http://orcid.org/0000-0002-3545-2811

## Ethics

Protocols for animal experiments were approved by the Committee on the Use of Live Animals in Teaching and Research (CULATR No. 3755-15), University of Hong Kong.

## Decision letter and Author response

Decision letter https://doi.org/10.7554/eLife.72182.sa1
Author response https://doi.org/10.7554/eLife.72182.sa2

# Additional files

## Supplementary files

• Transparent reporting form

## Data availability

All data generated or analyzed during this study are included in the manuscript and supporting files. Source data files have been provided.

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
