## [Editor Report]

This paper is of interest to a broad audience of cell biologists, pharmacologists and researchers who work in metabolic diseases. The work provides substantial new insights into the mechanism of action for a plant derived pentacyclic triterpene called celastrol elastrol, in effectively reducing the high fat diet induced tissue hypertrophy in mouse liver and adipose. A series of compelling experiments depict the site of covalent inhibition of the ER stress sensor GRP78 as essential for the beneficial effects in-vivo, supporting the main conclusions.

---

## [Decision Letter]

**Decision letter after peer review:**

Thank you for submitting your article "Covalent inhibition of endoplasmic reticulum chaperone GRP78 disconnects the transduction of ER stress signals to inflammation and lipid accumulation in diet-induced obese mice" for consideration by *eLife*. Your article has been reviewed by 3 peer reviewers, and the evaluation has been overseen by a Reviewing Editor and Mone Zaidi as the Senior Editor. The following individuals involved in review of your submission have agreed to reveal their identity: Ming-Yuen Lee (Reviewer #1).

Essential revisions:

We list here two new experiments and suggestion points for the discussion that we think are essential to support the claims made. These are:

1) Information regarding the cytotoxicity associated with celestrol is needed. This information is important and was requested by all the reviewers.

2) Reduced GRP78 expression is only shown by immunofluorescence staining in resident macrophage cells of mice liver and adipose tissues by the celastrol treatment. As suggested by Reviewer 2, isolation of macrophages from tissues can be performed in order to quantify protein or mRNA levels.

3) Expand the discussion of other potential effects of celastrol, given that the effects are not likely to be exclusive to the modulation of ER stress.

4) Revise the nomenclature of genes and proteins.

*Reviewer #1 (Recommendations for the authors):*

Overall, this manuscript was clearly written and presented enough data to support the conclusion. The authors should address the following issues to enhance the quality of manuscript for publication in *eLife*.

Specific issues:

1. The authors should indicate the herbal source for celastrol.

2. The authors should provide more details for the toxicity of celastrol.

3. The authors mentioned that celastrol might target several signaling pathways. It would be important to provide the rationale for the identification of covalent conjugates for celastrol.

4. The authors should explain why RNA-sequencing was performed using Nano-celastrol-treated mice.

5. Figure 8, the images should be clearly labelled with celastrol-PEG4-alkyne or celastrol in the incubation with the cells.

6. Figure 8, the labeling of celastrol-PEG4-alkyne appeared to be more extensive than the immunostaining of GRP78. The authors should discuss the binding of celastrol to other cellular proteins.

*Reviewer #2 (Recommendations for the authors):*

1. Figure 1 and Figure 2. These results can be further strengthened by including the protein and/or mRNA expression of the upstream and downstream regulators of ER stress. This will ensure that the pathway under study is engaged in the experimental model and that it is being modulated by celastrol.

2. Figure 3. At least, a quantification of Figure 3 showing the changes in the relative number of immune cell infiltration and their respective GRP78 expression should be added.

3. Figure 4. Authors should discuss why the major regulators of canonical ER stress pathway are not in the list of differentially expressed genes.

4. Figure 5. Explaining the results depicted in figure 5C, authors mention that celastrol has led to a decrease in the protein levels of IRE1, ATF6 and GRP78 in the ER of macrophage cells, which is evident. But the data does not support their claim that GRP78 localization is affected by celastrol treatment. What is evident in the images is the level of GRP78. One obvious reason is relatively small size of RAW 264.7 macrophage cells. As a proof of concept, one possibility is to use giant mammalian cells like U2OS or H9C2 which are better suited for subcellular compartment studies.

5. An important experiment that the authors should consider is to show that how much of the effect of celastrol is through GRP78? Anti-inflammatory and lipolytic effects of celastrol should be studied after a genetic or siRNA mediated knock down of GRP78 under conditions of lipotoxicity and celastrol treatment.

6. Figure 5. Line 194- In the Results section describing Figure 5, authors write "As shown in Figure 5C, palmitate significantly increased the expression of GRP78, 195 IRE1α, ATF6, … " while no description of significance calculation is shown in the figure legend neither any graphs showing the results for that.

7. Figure 4 legend, line no. 1051 – HDF should be HFD.

8. Figure 9 – is there any cytotoxicity associated with celestrol? Authors should state the number of mice died in each group and if any was excluded from the final analysis.

9. Figure 5B – authors should include a description of the abbreviation used here in the figure legend. What does CL, FT, W1, W2, W3, W4 and E mean?

10. One suggestion is to add a one to two sentence short summary at the end of each result section. This will help to better understand the findings that have been presented.

11. Authors have relied on macrophage cells to show the binding and localization of celastrol with GRP78. This should also be reflected in the title of the manuscript.

12. A graphical abstract highlighting important findings of the study will help to gain more appreciation.

13. Labelling for molecular weight should be added to the western blot images, at least to the source data.

*Reviewer #3 (Recommendations for the authors):*

Page 7, Lines 112-114, 116-118, 122-125:

All the lines are describing the regulations of different genes expression in the disease model, but the formatted symbols included are all corresponding to the protein of the respective genes such as in line 112 "adipogenic genes (e.g., PPAR-γ, C/EBP-α, aP2)", PPAR-γ refer to the protein and it is better to change it to Pparg to avoid misunderstanding.

Page 17, Lines 303-304, 306-309:

It is describing the gene expression again but all the symbols are corresponding the respective proteins and should be in italics.

Page 52, Figure 2, (A)(B)(C):

The figures are describing the fold changes of gene expression of different tissues after treatment but all the symbols in the figure are referring the proteins not the gene symbols

Page 52, Lines 1031, 1032:

In line 1031," in epididymal adipose tissues." But in line 1032 "total RNAs were extracted from liver" which is confusing and the source of RNA extracted is unclear.

Page 55, Figure 5 (A):

It is describing the effect of celastrol on ER stress in RAW264.7 macrophages. Maybe it is better to show that the concentrations used for celastrol doesn't induce any cytotoxicity in RAW264.7 after incubation for 8 hours.

Page 56, Figure 6 (B):

It is better to have some explanation on the annotations of the symbols on graph and tell the audience what kind of samples load in each lane.

Page 59, Figure 9 (E)(F):

It is better to also include the graph of IPGTT and ITT tests with the blood glucose level plot against time instead of showing the Area under curve data.

---

## [Author Response]

Essential revisions:Reviewer #1 (Recommendations for the authors):Overall, this manuscript was clearly written and presented enough data to support the conclusion. The authors should address the following issues to enhance the quality of manuscript for publication in eLife.Specific issues:1. The authors should indicate the herbal source for celastrol.

Thank you very much for your suggestion. We have indicated the herbal source for celastrol in the lines 66-67 and the commercial source in Methods. For the present study, we purchased celastrol with the purity of over 98% (HPLC) from Nanjing Spring and Autumn Biological Engineering Co., Ltd. (Nanjing, China).

2. The authors should provide more details for the toxicity of celastrol.

Thank you very much for your suggestion. We fully agree with the reviewer on the importance of addressing the toxicity of celastrol. Firstly, we have previously determined the effect of celastrol on liver by evaluating the levels of serum AST and ALT in DIO mice following the treatment with celastrol at the dose of 5mg/kg/day p.o. for 21 days. As shown in Author response image 1, celastrol at 5 mg/kg/day effectively suppressed the HFD-induced hepatotoxicity and did not exert further toxicity to the DIO mice.

**Author response image 1. sa2fig1:** 

Secondly, we examined the effects of celastrol on tissue integrity in the major organs and animal behaviors following 21-day treatment. As shown in the results of hematoxylin and eosin (HandE) staining, celastrol at the dose of 7.5 mg/kg/day, p.o. for 21 days did not alter on the integrity of the major mouse organs including the liver, heart, spleen and kidney (Scale bar: 100μm) nor showed any sign of intoxication.

Thirdly, Liu et al. previously evaluated the long-term toxicity of celastrol and did not find evident toxic effect in wild-type mice after i.p. administration of 100 μg/kg/day for 195 and 216 days (*Liu* J et al. Cell, *2015* May 21; 161(5):999-1011). However, celastrol is a highly hydrophobic drug and may be accumulated in the gastrointestinal tract. High gastrointestinal retention may affect the integrity of gastrointestinal tract. We have developed nanoparticle delivery system to resolve the gastrointestinal retention of celastrol (Zhao J et al., J Control Release. 2019 Sep 28;310:188-197).

3. The authors mentioned that celastrol might target several signaling pathways. It would be important to provide the rationale for the identification of covalent conjugates for celastrol.

Thank you very much for your suggestion. Many previous studies have suggested that celastrol could regulate different signaling pathways (Kannaiyan R, et al. Molecular targets of celastrol derived from Thunder of God Vine: potential role in the treatment of inflammatory disorders and cancer. Cancer Lett. 2011 Apr 1;303(1):9-20; Chen SR et al. A Mechanistic Overview of Triptolide and Celastrol, Natural Products from Tripterygium wilfordii Hook F. Front Pharmacol. 2018 Feb 14;9:104). However, most of celastrol-targeting signaling pathways appear to be the downstream signaling pathways while little is known about the primary molecular targets. Celastrol bears a unique chemically reactive quinone methide and may likely bind to proteins via covalent conjugation. Along the effort to identify the primary target proteins, we designed N-propargyl celastrol amide as a small molecule probe for affinity pulldown of celastrol-modified proteins. We are pleased to report our success in the identification of GRP78 as a major primary target for celastrol.

4. The authors should explain why RNA-sequencing was performed using Nano-celastrol-treated mice.

Thank you very much for your question. We have been working on two pharmacological aspects, bioavailability and molecular mechanisms, towards the development of clinically useful celastrol preparations. As mentioned in the manuscript, our previous study reported the preparation of Nano-celastrol by entrapping celastrol in PEG-PCL nanomicelles for safe control of body weight (Zhao et al. 2019). Nano-celastrol showed competitive efficacy and less gastrointestinal retention relative to free celastrol. As for transcriptomic profiles, a previous study reported the global transcriptional response at genes and enhancers to the natural medicinal compound celastrol (Dukler N, et al. Nascent RNA sequencing reveals a dynamic global transcriptional response at genes and enhancers to the natural medicinal compound celastrol. Genome Res. 2017 Nov;27(11):1816-1829). Thus, we performed RNA-sequencing to profile the transcriptional response to Nano-celastrol in mice. Indeed, our RNA-Seq profiling revealed the effects of Nano-celastrol on the biomarkers of ER stress, inflammation, lipolysis, lipogenesis and thermogenesis. Importantly, such information well supported the conclusion in the present manuscript.

5. Figure 8, the images should be clearly labelled with celastrol-PEG4-alkyne or celastrol in the incubation with the cells.

Thanks for your suggestion. We have properly labelled the images in Figure 8.

6. Figure 8, the labeling of celastrol-PEG4-alkyne appeared to be more extensive than the immunostaining of GRP78. The authors should discuss the binding of celastrol to other cellular proteins.

Thanks for your comments. The extensive labelling with celastrol-PEG4-alkyne may be attributed to three factors: (1) the different brightness and sensitivities of green and red fluorophores, (2) non-specific binding and (3) actual binding of celastrol-PEG4-alkyne to other minor proteins. We have updated such discussion in the revised manuscript.

Reviewer #2 (Recommendations for the authors):1. Figure 1 and Figure 2. These results can be further strengthened by including the protein and/or mRNA expression of the upstream and downstream regulators of ER stress. This will ensure that the pathway under study is engaged in the experimental model and that it is being modulated by celastrol.

Thank you very much for your suggestion. We fully agree with the reviewer on the importance to determine the protein and/or mRNA expression of the upstream and downstream regulators of ER stress. We performed new experiments and determined the effects of celastrol on the expression of GRP78 and related ER stress signaling proteins in livers and epididymal fat pads by Western blotting. We have added the new results into the revised Figure 3.

2. Figure 3. At least, a quantification of Figure 3 showing the changes in the relative number of immune cell infiltration and their respective GRP78 expression should be added.

Thank you very much. We highly respect you for your comments. We are aware of the technical inaccuracy in the quantification of the fluorescence signals in tissues, especially for enumerating cells. We performed new experiments to analyze the protein levels of GRP78 and related ER stress signaling biomarkers in livers and epididymal fat pads by Western blotting. The blots were quantified by a densitometric method. Thus, we included the new results in the revised Figure 3.

3. Figure 4. Authors should discuss why the major regulators of canonical ER stress pathway are not in the list of differentially expressed genes.

Thank you very much for your question. Based on standard bioinformatics analysis, the list of differentially expressed genes recruited the genes with the change of > 2-fold. On the other hand, the major regulators of canonical ER stress pathway are mainly the sensors for ER stress. The minimal changes of ER stress regulators may result in the sufficient effects on the biomarkers of inflammation, lipolysis, lipogenesis and thermogenesis.

4. Figure 5. Explaining the results depicted in figure 5C, authors mention that celastrol has led to a decrease in the protein levels of IRE1, ATF6 and GRP78 in the ER of macrophage cells, which is evident. But the data does not support their claim that GRP78 localization is affected by celastrol treatment. What is evident in the images is the level of GRP78. One obvious reason is relatively small size of RAW 264.7 macrophage cells. As a proof of concept, one possibility is to use giant mammalian cells like U2OS or H9C2 which are better suited for subcellular compartment studies.

Thank you very much for your comments. We fully agree with the reviewer on the expression and localization of GRP78 in macrophages. Macrophages are the major cell type for handling the dietary lipids, especially in the transformation into foam cells, and thereby frequently used as the cell model for studying inflammation and lipid metabolism. As for the broad distribution, GRP78 may be localized in the ER compartment as its primary localization or transported to the cell surface under stress (Arap MA et al. Cell surface expression of the stress response chaperone GRP78 enables tumor targeting by circulating ligands. Cancer Cell. 2004 Sep;6(3):275-84). Per your suggestion, it would be interesting to study the effects of fatty acids on the expression and localization of GRP78 in giant mammalian cells like U2OS or H9C2 in a separated study. Indeed, we have published several papers with H9C2 in the context of myocardial infarction.

5. An important experiment that the authors should consider is to show that how much of the effect of celastrol is through GRP78? Anti-inflammatory and lipolytic effects of celastrol should be studied after a genetic or siRNA mediated knock down of GRP78 under conditions of lipotoxicity and celastrol treatment.

Thanks for your comments. The present study was designed to identify the celastrol-modified proteins. Indeed, we successfully identified GRP78 as a major target protein for celastrol and proved that celastrol could attenuate the chaperone activity of GRP78. Based on careful literature review, it has been well-established that ER stress promotes lipogenesis and inflammation (Kim JY et al. ER Stress Drives Lipogenesis and Steatohepatitis via Caspase-2 Activation of S1P. Cell. 2018 Sep 20;175(1):133-145.e15; Lee JS et al. Pharmacological ER stress promotes hepatic lipogenesis and lipid droplet formation. Am J Transl Res. 2012;4(1):102-13. Epub 2012 Jan 6). The present study demonstrated that celastrol suppressed the expression of ER stress mediators, inflammatory and lipogenic genes while promoted lipolytic and thermogenic genes. Collectively, it is reasonable to propose an important role for the formation of celastrol-GRP78 conjugate. Again, we highly appreciate the reviewer for your constructive suggestions. Indeed, we designed another study to further examine the anti-inflammatory and lipolytic effects of celastrol and celastrol derivatives using different technologies including AAV-mediated knockdown of GRP78 in cell models and animal models.

6. Figure 5. Line 194- In the Results section describing Figure 5, authors write "As shown in Figure 5C, palmitate significantly increased the expression of GRP78, 195 IRE1α, ATF6, … " while no description of significance calculation is shown in the figure legend neither any graphs showing the results for that.

Thank you very much for your comments. The images in Figure 5C visually showed that palmitate markedly increased the expression of GRP78, IRE1α, ATF6. Per your suggestion, we rephrased the statement in the revised.

7. Figure 4 legend, line no. 1051 – HDF should be HFD.

Thank you very much for your comments. We corrected the typos in the text.

8. Figure 9 – is there any cytotoxicity associated with celestrol? Authors should state the number of mice died in each group and if any was excluded from the final analysis.

See response to reviewer 1, point 2.

9. Figure 5B – authors should include a description of the abbreviation used here in the figure legend. What does CL, FT, W1, W2, W3, W4 and E mean?

Thank you very much for your question. Per your suggestion, these abbreviations have been added in the figure legend for you to review. Briefly, CL: Cell lysate; FT: Flow-through; W1: First wash; W2: Second wash; W3: Third wash; W4: Fourth wash; E: Elution.

10. One suggestion is to add a one to two sentence short summary at the end of each result section. This will help to better understand the findings that have been presented.

Thank you very much for your suggestion. We have added short summary at the end of each result section.

11. Authors have relied on macrophage cells to show the binding and localization of celastrol with GRP78. This should also be reflected in the title of the manuscript.

Thank you very much for your suggestion. GRP78 is an abundant and ubiquitous protein and governs a common ER stress signaling pathway. In view of the role of macrophages in the metabolism of lipids, we selected macrophages as cell model to demonstrate a general pharmacological phenomenon. Moreover, animal experiments showed overall response to HFD and celastrol treatment. Therefore, it may be better to keep the current title.

12. A graphical abstract highlighting important findings of the study will help to gain more appreciation.

Thank you very much for your comments. Per your suggestion, we have added the graphical summary with highlights as a new Figure 10 into the revised manuscript for you to review.

13. Labelling for molecular weight should be added to the western blot images, at least to the source data.

Thank you very much for your comments. We have added the molecular weight of proteins to the Western blot images in Figure 3 and Figure 5 for you to review.

Reviewer #3 (Recommendations for the authors):Page 7, Lines 112-114, 116-118, 122-125:All the lines are describing the regulations of different genes expression in the disease model, but the formatted symbols included are all corresponding to the protein of the respective genes such as in line 112 "adipogenic genes (e.g., PPAR-γ, C/EBP-α, aP2)", PPAR-γ refer to the protein and it is better to change it to Pparg to avoid misunderstanding.

Thank you very much for your suggestion. We have corrected the symbols of different genes accordingly.

Page 17, Lines 303-304, 306-309:It is describing the gene expression again but all the symbols are corresponding the respective proteins and should be in italics.

Thank you very much for your suggestion. We have corrected the symbols of different genes accordingly.

Page 52, Figure 2, (A)(B)(C):The figures are describing the fold changes of gene expression of different tissues after treatment but all the symbols in the figure are referring the proteins not the gene symbols

Thank you very much for your suggestion. We have corrected the symbols of different genes accordingly in Figure 2.

Page 52, Lines 1031, 1032:In line 1031," in epididymal adipose tissues." But in line 1032 "total RNAs were extracted from liver" which is confusing and the source of RNA extracted is unclear.

Thank you very much for your suggestion. We have corrected the source of RNA extracted into epididymal adipose tissues in the figure legend.

Page 55, Figure 5 (A):It is describing the effect of celastrol on ER stress in RAW264.7 macrophages. Maybe it is better to show that the concentrations used for celastrol doesn't induce any cytotoxicity in RAW264.7 after incubation for 8 hours.

Thank you very much for your comments. Our previous study has already evaluated the effect of celastrol on the cell viability in RAW264.7 cells over a period of 48 h (Luo D et al. Natural product celastrol suppressed macrophage M1 polarization against inflammation in diet-induced obese mice via regulating Nrf2/HO-1, MAP kinase and NF-κB pathways. Aging (Albany NY). 2017 Oct 16;9(10):2069-2082). We found that celastrol could exhibit detectable cytotoxicity when the concentration was raised up to 1.25 μM (*p* < 0.01). Celastrol eventually exhibited the IC_50_ value of 1.69 μM against the growth of RAW264.7 cells. We thereby treated RAW264.7 cells with celastrol at the concentrations below 1.25 μM for other experiments. Therefore, 1μM of celastrol wouldn't induce any cytotoxicity in RAW264.7 after incubation for 8 hours. Please refer to Author response image 2.

Page 56, Figure 6 (B):It is better to have some explanation on the annotations of the symbols on graph and tell the audience what kind of samples load in each lane.

Thank you very much for your comments. The abbreviations have been explained in the figure legend for you to review. Briefly, CL: Cell lysate; FT: Flow-through; W1: First wash, W2: Second wash; W3: Third wash; W4: Fourth wash; E: Elution.

Page 59, Figure 9 (E)(F):It is better to also include the graph of IPGTT and ITT tests with the blood glucose level plot against time instead of showing the Area under curve data.

Thank you very much for your comments. We included the quantitative analysis as a part of Figure 9 due to the consideration of easy understanding, visual tidiness and consistent format. To facilitate your review, we have made the graph of IPGTT and ITT tests with the blood glucose level plot against time in Author response image 3.

**Author response image 3. sa2fig3:**